# Identification and Estimation of Joint Probabilities of Potential Outcomes in Observational Studies with Covariate Information

**Ryusei Shingaki    Manabu Kuroki**
Graduate School of Engineering Science, Yokohama National University
shingaki-ryusei-kw@ynu.jp    kuroki-manabu-zm@ynu.ac.jp

## Abstract

The joint probabilities of potential outcomes are fundamental components of causal inference in the sense that (i) if they are identifiable, then the causal risk is also identifiable, but not vise versa (Pearl, 2009; Tian and Pearl, 2000) and (ii) they enable us to evaluate the probabilistic aspects of "necessity", "sufficiency", and "necessity and sufficiency", which are important concepts of successful explanation (Watson, et al., 2020). However, because they are not identifiable without any assumptions, various assumptions have been utilized to evaluate the joint probabilities of potential outcomes, e.g., the assumption of monotonicity (Pearl, 2009; Tian and Pearl, 2000), the independence between potential outcomes (Robins and Richardson, 2011), the condition of gain equality (Li and Pearl, 2019), and the specific functional relationships between cause and effect (Pearl, 2009). Unlike existing identification conditions, in order to evaluate the joint probabilities of potential outcomes without such assumptions, this paper proposes two types of novel identification conditions using covariate information. In addition, when the joint probabilities of potential outcomes are identifiable through the proposed conditions, the estimation problem of the joint probabilities of potential outcomes reduces to that of singular models and thus they can not be evaluated by standard statistical estimation methods. To solve the problem, this paper proposes a new statistical estimation method based on the augmented Lagrangian method and shows the asymptotic normality of the proposed estimators. Given space constraints, the proofs, the details on the statistical estimation method, some numerical experiments, and the case study are provided in the supplementary material.

## 1 Introduction

### 1.1 Practical Background

In practical science, it is crucial to evaluate the likelihood of one event causing another event. For example, epidemiologists pay attention to determining the likelihood of a particular exposure being the cause of a particular disease. In order to assess such likelihood from statistical data, the probabilities of causation have been developed, which can be divided into "necessary causation", "sufficient causation", and "necessary-and-sufficient causation". In addition, recently, in the field of explainable artificial intelligence (XAI), it has been pointed out that "necessity", "sufficiency", and "necessity and suffciency" are the important concepts of successful explanation and the probabilities of causation play an important role

35th Conference on Neural Information Processing Systems (NeurIPS 2021).

in evaluating these concepts from the probabilistic aspects [10, 22, 35]. For the last decades, the probabilities of causation received increasing attention in the fields of epidemiology, risk analysis, legal reasoning, artificial intelligence, and policy analysis [e.g., 4–8, 10, 11, 22, 26, 29, 33, 35].

## 1.2 "Necessity", "Sufficiency", and "Necessity and Sufficiency"

Necessary cause is often used in the tort-law liability. When a 60-year-old man who worked in an asbestos textile plant develops lung cancer, we are interested in the probability that asbestos had caused it. This quantity to be estimated has been commonly referred to as "attributable fraction" or "assigned share" [4, 19]. It measures how necessary the exposure is for the development of the disease. Pearl [24] defined it as "Probability of Necessity (PN)", which means "the probability that disease would not have occurred in the absence of exposure, given that disease and exposure did in fact occur". The estimation of PN is important to provide a scientific basis for compensation in tort suits.

Sufficient cause is often used in health risk assessment. In practice, it is important for public policy makers to predict how a health risk factor would endanger a healthy population [15, 16, 21]. The measure for it has been referred to as "susceptibility", which measures how sufficient a risk factor is to make an individual develop a disease. Pearl [24] defined the measure of such prediction as "Probability of Sufficiency (PS)", which means "the probability that a healthy unexposed individual would have developed the disease had he been exposed". The assessment of PS helps public policy makers to make or revise regulations in order to minimize the occurrence of diseases.

With the definitions of PN and PS, it would be a natural extension to inquire for the "Probability of Necessity and Sufficiency (PNS)", which shows how likely an individual is to be affected by both ways [24]. PNS measures the probability that an exposure is a necessary and sufficient cause for a disease, that is, an exposure is the actual cause of a disease. Then, different from previous works where "probability of causation" is only referred to PN [e.g., 4, 7], Pearl [24] defined three types of "probability of causation", PN, PS, and PNS.

Pearl [26] and Tian and Pearl [33] developed formal semantics for the probabilities of causation based on the structural models of counterfactuals. They presented the formal definitions of the probability of necessity (PN), the probability of sufficiency (PS), and the probability of necessity and sufficiency (PNS). These probabilities of causation are formulated based on the joint probabilities of two potential outcomes. Since one cannot simultaneously observe the results of the same subjects exposed and unexposed in reality, even successful randomized experiments have failed to identify these quantities [26, pp.284–285]. Tian and Pearl [33] addressed the problem by showing how to bound these quantities using data obtained from experimental and observational studies. Their bounds are sharp under minimal assumptions concerning the data generating process. Kuroki and Cai [17] derived narrower bounds of the probabilities of causation than Tian-Pearl's bounds using covariate information. Recently, Galhotra et al. [10] generalized Tian-Pearl's bounds from binary to arbitrary variables in the context of explainable artificial intelligence (XAI). However, it has been often pointed out that these existing bounds are too wide to evaluate the probabilities of causation. To solve the problem, Tian and Pearl [33] and Galhotra et al. [10] also stated that the probabilities of causation are identifiable if the monotonicity can be assumed and the causal risks are identifiable and Pearl [26] showed that the specific functional relationships between cause and effect lead to the identification of the probabilities of causation.

## 1.3 Contribution

The aim of this paper is to discuss the identification and estimation problems of the joint probabilities of potential outcomes, which provides a general class including the probabilities of causation. The joint probabilities of potential outcomes are the building-blocks of causal inference, because the causal risk is also identifiable if they are identifiable, but not vise versa. In the context of the natural direct and indirect effects [25], under the assumption of no unmeasured confounding, Robins and Richardson [30] stated that the joint probabilities of potential outcomes are identifiable if (i) two potential outcomes are independent or (ii) one potential outcome is deterministically formulated as the function of the other potential

outcome. In addition, in the context of unit selection problems, Li and Pearl [20] showed that the linear combination of the joint probabilities of potential outcomes is identifiable under the gain equality. These existing researches show that the evaluation of the joint probabilities of potential outcomes play an important role in solving various problems of causal inference. Although considerable efforts have been devoted toward establishing the identifiability criteria and solving the confounding problem of causal risks, there has been limited discussion on how to identify the joint probabilities of potential outcomes when the present assumptions are violated.

To solve the problem, this paper proposes two types of novel identification conditions for the joint probabilities of potential outcomes without existing assumptions — one uses both causal risks and one covariate (Theorem 1) and the other uses two covariates without causal risks (Theorem 2). The results of this paper also provide a counterexample to "sensitivity to the generative process" [26, pp. 284–285], in the sense that it is not required to specify the functional relationships that connect a cause and an effect. In addition, unlike the effect restoration proposed by Kuroki and Pearl [18], the assumption of the given number of categories of an unmeasured confounder is not required. Furthermore, from Theorem 2, it is remarkable that there are some situations where the causal risks are identifiable even if the existing identification conditions do not hold.

When the joint probabilities of potential outcomes are identifiable through the proposed conditions, the estimation problem of the joint probabilities of potential outcomes reduces to that of singular models and thus they can not be evaluated by standard statistical estimation methods. To solve the problem, this paper proposes a new statistical method based on the augmented Lagrangian method [3] and shows the asymptotic normality of the proposed estimators. Given space constraints, the proofs, the details on the statistical estimation method, some numerical experiments, and the case study are provided in the supplementary material.

## 2  Preliminary

This section introduces the potential outcomes used to discuss our problems. For simplicity, we consider the case where an exposure variable $X \in \{x_1, x_0\}$ ($x_1$ is exposed; $x_0$ is unexposed) and an outcome variable $Y \in \{y_1, y_0\}$ ($y_1$ is disease; $y_0$ is non-disease) are dichotomous. In addition, let $p(X = x, Y = y) = p(x, y)$, $p(Y = y \mid X = x) = p(y \mid x)$, and $p(X = x) = p(x)$ be the joint probability of $(X, Y) = (x, y)$, the conditional probability of $Y = y$ given $X = x$, and the marginal probability of $X = x$ for $x \in \{x_1, x_0\}$ and $y \in \{y_1, y_0\}$, respectively. Similar notation is used for other probabilities. Here, in this paper, we will assume that readers are familiar with the basic theory of causal inference [13, 26]. Especially, for the graph-theoretic terminology and the basic theory of structural causal models used in this paper, we refer the readers to Pearl [26].

In principle, for $x \in \{x_1, x_0\}$, the $i$-th of the $N$ subjects has a potential outcome $Y_x(i)$ that would have resulted if $X$ had been $x$, denoted as $X(i) = x$. Here, note that the subject ensures a deterministic relationship between two variables $X$ and $Y$ in the semantics of structural causal models [26]. In addition, the present paper assumes the stable unit treatment value assumption, which can be summarized as follows: (i) the exposure status of any subject does not affect the outcomes of the other subjects (no interference) and (ii) the exposures of all subjects are comparable (no variation in exposure). Thus, when the $N$ subjects in the study are considered as random samples from the population of interest, $X(i)$ and $Y_x(i)$ are referred to as the values of random variables $X$ and $Y_x$ respectively, and thus the causal risk of $X = x$ on $Y = y$ is defined as $p(Y_x = y)$. Similar notation is used for other potential outcomes.

For the $i$-th subject, the potential outcome $Y_x$ is observed only if $X$ is $x$ . This property is called the consistency [13, 26, 28], which is formulated as $X(i) = x \Longrightarrow Y_x(i) = Y(i)$. When a randomized experiment is conducted and compliance is perfect, since $X$ is independent of $(Y_{x_0}, Y_{x_1})$, the causal risk is identifiable and is given by $p(Y_x = y) = p(y \mid x)$. Here, "identifiable" means that the causal quantities, such as $p(Y_x = y)$, can be estimated consistently from a joint probability of observed variables. In contrast, when it is difficult to conduct

a randomized experiment and only observational data are available, we can evaluate the causal risk according to the conditionally-ignorable-treatment-assignment condition [31], or graphically, the back-door criterion [26]. In other words, for the exposure $X$, if there exists such a set $\boldsymbol{S}$ of observed covariates that $X$ is conditionally independent of $(Y_{x_0}, Y_{x_1})$ given $\boldsymbol{S}$, we can say that treatment assignment is conditionally ignorable given $\boldsymbol{S}$. Then, the causal effects are estimable using $\boldsymbol{S}$ as $p(Y_x = y) = E_s[p(y \,|\, x, \boldsymbol{S})]$. Here, $E_s[p(y \,|\, x, \boldsymbol{S})]$ is the expectation of $p(y \,|\, x, \boldsymbol{S})$ regarding $\boldsymbol{S}$. Although there are other identification conditions that can be used to solve our problem [e.g., 26, 34], this paper does not cover them due to space constraints.

Following Pearl [26], we define three probabilities of causation, that is, the probability of necessity (PN), the probability of sufficiency (PS), and the probability of necessity and sufficiency (PNS). PN stands for the probability $p(Y_{x_0} = y_0 \,|\, X = x_1, Y = y_1)$ that a disease would not have occurred in the absence of an exposure, given that the exposure and the disease occurred. PS stands for the probability $p(Y_{x_1} = y_1 \,|\, X = x_0, Y = y_0)$ that a healthy unexposed subject would have contracted the disease upon exposure. PNS stands for the probability $p(Y_{x_1} = y_1, Y_{x_0} = y_0)$ that an exposure is a necessary and sufficient cause for a disease. Since PN, PS, and PNS involve the joint probabilities of potential outcomes, even under successful randomized experiments, they cannot be estimated from the observed data without additional assumptions [26].

## 3 Identification

### 3.1 Case 1: Identification based on one covariate with causal risks

According to Pearl [26], letting $\boldsymbol{U}$ be the set of all discrete and continuous covariates that could affect $X$ and $Y$, both observed and unobserved, we consider the problem of evaluating the joint probabilities of potential outcomes based on the directed graph shown in Figure 1a. Here, a covariate $Z$, in Figure 1a, is measured as a proxy variable of $\boldsymbol{U}$. Note that $Z$ can be a set of discrete and/or continuous variables. In Figure 1a, the directed edge from $X$ to $Y$ indicates that $X$ may affect $Y$. In addition, the absence of a directed edge from $Y$ to $X$ indicates that $Y$ cannot be a cause of $X$, and the directed path from $\boldsymbol{U}$ to $Y$ through $X$ indicates that some elements of $\boldsymbol{U}$ may effect $Y$ mediated by $X$. Figure 1a also graphically represents the data generating process

$$Y = g_y(X, \boldsymbol{U}, \epsilon_y), X = g_x(\boldsymbol{U}, \epsilon_x), Z = g_z(\boldsymbol{U}, \epsilon_z), \tag{1}$$

where $\epsilon_x$, $\epsilon_y$, and $\epsilon_y$ are independent random disturbances and they are also independent of $\boldsymbol{U}$. When the structural equation models, such as equation (1), are used to represent the data generating process, the corresponding graph shown in Figure 1a is called a causal diagram.

The situation such as Figure 1a is also discussed in Kuroki and Pearl [18] and Pearl [27]. However, in this paper, unlike Kuroki and Pearl [18] and Pearl [27], $\boldsymbol{U}$ can include uncertain number of all discrete and continuous covariates that influence the way a subject responds to exposures. Thus, in many situations, it is reasonable to assume the existence of a covariate $Z$ that is independent of $\{X, Y\}$ given $\boldsymbol{U}$. Here, note that the independence assumptions between two observed variables would be affected by partitioning the states of $\boldsymbol{U} \cup \{\epsilon_x, \epsilon_y \, \epsilon_z\}$.

In the situation shown in Figure 1a, irrespective of the complexity of $\boldsymbol{U} \cup \{\epsilon_x, \epsilon_y, \epsilon_z\}$, the impact of $\boldsymbol{U}$ on $Y$ remains restricted to the modification of the functional relationships between $X$ and $Y$. This yields four functions for two dichotomous variables $X$ and $Y$, and thus the value taken by $\boldsymbol{U} \cup \{\epsilon_x, \epsilon_y, \epsilon_z\}$ selects one of these four functions [26]. Considering these observations, according to Rothman et al. [32, p. 59], the states of $\boldsymbol{U} \cup \{\epsilon_x, \epsilon_y, \epsilon_z\}$ are divided into the following four potential outcome types:

$u_1 = (Y_{x_1} = y_1, Y_{x_0} = y_1)$ represents a "doomed" situation where exposure is irrelevant because disease occurs with or without exposure.

$u_2 = (Y_{x_1} = y_1, Y_{x_0} = y_0)$ represents a "causative" situation where disease occurs if and only if subjects are exposed.

$u_3 = (Y_{x_1} = y_0, Y_{x_0} = y_1)$ represents a "preventive" situation where disease occurs if and only if subjects are unexposed.

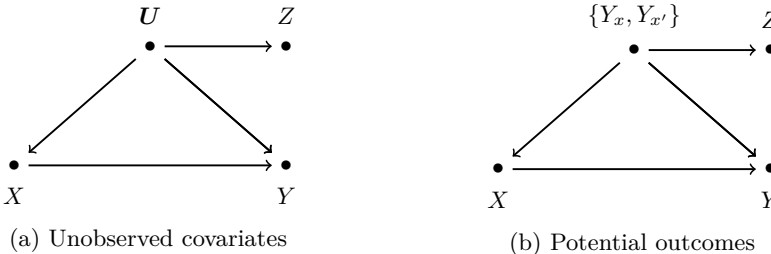

(a) Unobserved covariates     (b) Potential outcomes

Figure 1: Graphical representation in Case 1.

$u_4 = (Y_{x_1} = y_0, Y_{x_0} = y_0)$ represents an "immune" situation where exposure is irrelevant because disease does not occur, with or without exposure.

According to this partition of the states of $\boldsymbol{U} \cup \{\epsilon_x, \epsilon_y, \epsilon_z\}$, it is re-defined as a variable $U$ taking a value $u$ ($u \in \{u_1, u_2, u_3, u_4\}$). Then, similar to the setting of the non-compliance problem by Pearl [26, ch.8], throughout the paper, we assume that the recursive factorization of the positive joint probability of $X$, $Y$, and $Z$, $p(x, y, z)$, is given as

$$p(x, y, z) = \sum_{i=1}^{4} p(y \mid x, z, u_i) p(x \mid u_i) p(z \mid u_i) p(u_i)$$

according to Figure 1b. It is also assumed that other joint probabilities of $X$, $Y$, and $Z$ can be rewritten in a similar way. Then, this factorization implies that $X$ is conditionally independent of $Z$, given $U$, denoted as $X \perp\!\!\!\perp Z \mid U$, which is used in the context of the impact evaluation problem of the program [23]. In addition, from the consistency property, we have

$$\begin{aligned}
p(x_1, y_1, z, u_i) &= p(x_1, z, u_i), & p(x_1, y_0, z, u_i) &= 0, & i &= 1, 2, \\
p(x_1, y_0, z, u_i) &= p(x_1, z, u_i), & p(x_1, y_1, z, u_i) &= 0, & i &= 3, 4, \\
p(x_0, y_1, z, u_i) &= p(x_0, z, u_i), & p(x_0, y_0, z, u_i) &= 0, & i &= 1, 3, \\
p(x_0, y_0, z, u_i) &= p(x_0, z, u_i), & p(x_0, y_1, z, u_i) &= 0, & i &= 2, 4,
\end{aligned}$$

Thus, for example, $p(x_1, y_1, z)$, $p(z)$ and $p(Y_{x_1} = y_1, z)$ can be rewritten as

$$p(x_1, y_1, z) = \sum_{i=1}^{2} p(x_1 \mid u_i) p(z \mid u_i) p(u_i), \quad p(z) = \sum_{i=1}^{4} p(z \mid u_i) p(u_i)$$

$$p(Y_{x_1} = y_1, z) = p(Y_{x_1} = y_1 \mid z) p(z).$$

Additionally, the other joint and marginal probabilities of $X$, $Y$, and $Z$ can be rewritten in a similar way. Then, when $Z$ is a variable with the number of values $k \geq 4$, say $z_1, \ldots, z_4$, letting

$$P = \begin{pmatrix}
1 & p(z_1) & p(z_2) & p(z_3) \\
p(x_1, y_1) & p(x_1, y_1, z_1) & p(x_1, y_1, z_2) & p(x_1, y_1, z_3) \\
p(x_1, y_0) & p(x_1, y_0, z_1) & p(x_1, y_0, z_2) & p(x_1, y_0, z_3) \\
p(x_0, y_1) & p(x_0, y_1, z_1) & p(x_0, y_1, z_2) & p(x_0, y_1, z_3)
\end{pmatrix}, \tag{2}$$

$$Q = \begin{pmatrix}
1 & p(z_1) & p(z_2) & p(z_3) \\
p(Y_{x_1} = y_1) & p(Y_{x_1} = y_1, z_1) & p(Y_{x_1} = y_1, z_2) & p(Y_{x_1} = y_1, z_3) \\
p(Y_{x_0} = y_1) & p(Y_{x_0} = y_1, z_1) & p(Y_{x_0} = y_1, z_2) & p(Y_{x_0} = y_1, z_3)
\end{pmatrix}, \tag{3}$$

we derive the following theorem:

**Theorem 1** *Letting $Z$ be a variable with the number of values $k \geq 4$, say $z_1, \ldots, z_4$, $p(Y_{x_1} = y, Y_{x_0} = y')$ and $p(Y_{x_1} = y, Y_{x_0} = y' \mid x)$ are identifiable ($y, y' \in \{y_1, y_0\}; x \in \{x_1, x_0\}$) if the following conditions are satisfied:*

**Condition 1** *$p(Y_{x_1} = y_1 \mid z)$, $p(Y_{x_0} = y_1 \mid z)$, $p(x_1, y_1, z)$, $p(x_0, y_1, z)$, $p(x_1, y_0, z)$, and $p(x_0, y_0, z)$ are available for $z \in \{z_1, \ldots, z_4\}$.*

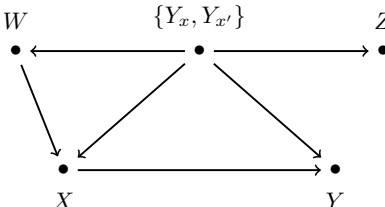

Figure 2: Graphical representation in Case 2

**Condition 2** $\{X, Y\} \perp\!\!\!\perp Z \mid U$ *holds.*

**Condition 3** *P is invertible and $QP^{-1}$ consists of four distinct non-zero column vectors.*

The proof is given in the supplementary material. Theorem 1 shows that PN, PS and PNS are identifiable. In addition, the causal risk is also identifiable from Theorem 1.

Here, we would like to state some remarks. First, the violation of the assumptions can be testable: as seen from the supplementary material, if the simultaneous linear equation (A.12) in the supplementary material has no solutions, then at least one of the conditions does not hold. Second, although Condition 1 seems to require that some distributions are available, it is not the case. Theorem 1 is applicable with a single observed distribution, say, $p(x, y, z, w)$ if both $p(Y_{x_1} = y \mid z)$ and $p(Y_{x_0} = y \mid z)$ are identifiable from the distribution. For example, regarding $p(x, y, z, w)$, when $\{Z, W\}$ satisfies the back-door criterion relative to $(X, Y)$, $p(Y_x = y \mid z)$ is identifiable and is given by

$$p(Y_x = y \mid z) = \sum_w p(y \mid x, z, w) p(w \mid z). \tag{4}$$

Second, Condition 3 implies that all potential outcome types exist and thus the monotonicity assumption (e.g. no-prevention assumption; $p(u_3) = 0$) does not hold. In addition, it implies that $X$ is not independent of $U$, and thus $X$ is not randomly assigned. Thus, in Theorem 1, the causal risks are assumed to be estimable by the identification conditions in observational studies (such as the back- and front-door criteria [26]) or by external studies (e.g., pilot studies or scientific judgement). If we have prior knowledge that a certain potential outcome type does not exist in the situation discussed this section, that is, the monotonicity assumption, then the PN, PS, and PNS would be identifiable without the covariate $Z$ [33]. Theorem 1 demonstrates that, when we observe one covariate associated with potential outcomes and the causal risks are available, the joint probabilities of potential outcomes can be constructed from the covariate and causal risks; this can render the joint probabilities of potential outcomes identifiable without the monotonicity assumption.

### 3.2 Case 2: Identification based on two covariates without causal risks

We will tackle the more difficult problem of estimating the joint probabilities of potential outcomes without a prior knowledge of causal risks. We will show that, under certain conditions, the joint probabilities of potential outcomes can be evaluated using only covariates. To see this, consider a causal diagram shown in Figure 2, which is obtained by adding an observed variable $W$ to Figure 1b. Then, when we consider the positive joint probability of $X, Y, Z$, and $W$, from the consistency property, we have

$$p(x_1, y_1, z, w, u_i) = p(x_1, z, w, u_i), \quad p(x_1, y_0, z, w, u_i) = 0, \quad i = 1, 2,$$
$$p(x_1, y_0, z, w, u_i) = p(x_1, z, w, u_i), \quad p(x_1, y_1, z, w, u_i) = 0, \quad i = 3, 4,$$
$$p(x_0, y_1, z, w, u_i) = p(x_0, z, w, u_i), \quad p(x_0, y_0, z, w, u_i) = 0, \quad i = 1, 3,$$
$$p(x_0, y_0, z, w, u_i) = p(x_0, z, w, u_i), \quad p(x_0, y_1, z, w, u_i) = 0, \quad i = 2, 4,$$

for $z$ and $w$ taken by $Z$ and $W$, respectively. Thus, for example, $p(y_1, z, w \mid x_1)$ and $p(z, w \mid x_1)$ can be rewritten as

$$p(y_1, z, w | x_1) = \sum_{i=1}^{2} p(w|x_1, u_i) p(z|u_i) p(u_i|x_1), \quad p(z, w | x_1) = \sum_{i=1}^{4} p(w|x_1, u_i) p(z|u_i) p(u_i|x_1).$$

Additionally, the other joint and marginal probabilities of $X$, $Y$, $Z$, and $W$ can be rewritten in a similar way. Then, when $Z$ and $W$ are variables with the number of values $k \geq 4$, such as $z_1, \ldots, z_4$, for $Z$, and $w_1, \ldots, w_4$ for $W$, letting

$$P_{x_1} = \begin{pmatrix} 1 & p(z_1 \mid x_1) & p(z_2 \mid x_1) & p(z_3 \mid x_1) \\ p(w_1 \mid x_1) & p(z_1, w_1 \mid x_1) & p(z_2, w_1 \mid x_1) & p(z_3, w_1 \mid x_1) \\ p(w_2 \mid x_1) & p(z_1, w_2 \mid x_1) & p(z_2, w_2 \mid x_1) & p(z_3, w_2 \mid x_1) \\ p(w_3 \mid x_1) & p(z_1, w_3 \mid x_1) & p(z_2, w_3 \mid x_1) & p(z_3, w_3 \mid x_1) \end{pmatrix}, \tag{5}$$

$$P_{x_0} = \begin{pmatrix} 1 & p(z_1 \mid x_0) & p(z_2 \mid x_0) & p(z_3 \mid x_0) \\ p(w_1 \mid x_0) & p(z_1, w_1 \mid x_0) & p(z_2, w_1 \mid x_0) & p(z_3, w_1 \mid x_0) \\ p(w_2 \mid x_0) & p(z_1, w_2 \mid x_0) & p(z_2, w_2 \mid x_0) & p(z_3, w_2 \mid x_0) \\ p(w_3 \mid x_0) & p(z_1, w_3 \mid x_0) & p(z_2, w_3 \mid x_0) & p(z_3, w_3 \mid x_0) \end{pmatrix}, \tag{6}$$

we derive the following theorem:

**Theorem 2** *Letting $Z$ and $W$ be variables by taking $k$ ($\geq 4$) values, say $z_1, \ldots, z_4$, for $Z$, and $w_1, \ldots, w_4$ for $W$, $p(Y_{x_1} = y, Y_{x_0} = y')$ and $p(Y_{x_1} = y, Y_{x_0} = y' \mid x)$ are identifiable $(y, y' \in \{y_1, y_0\}; x \in \{x_1, x_0\})$ if the following conditions are satisfied:*

**Condition 4** *$p(x_1, y_1, z, w)$, $p(x_1, y_0, z, w)$, $p(x_0, y_1, z, w)$, and $p(x_0, y_0, z, w)$, are available for $z \in \{z_1, \ldots, z_4\}$ and $w \in \{w_1, \ldots, w_4\}$.*

**Condition 5** *Both $W \perp\!\!\!\perp Z \mid \{X, U\}$ and $X \perp\!\!\!\perp Z \mid U$ hold.*

**Condition 6** *Both $P_{x_1}$ and $P_{x_0}$ are invertible.*

The proof is given in the supplementary material. Although the probabilities comprising the unobserved variables are not fully identified in Kuroki and Pearl [18], Theorem 2 shows that the joint probabilities of potential outcomes $p(u_1), \ldots, p(u_4)$ are identifiable and the causal risks are also identifiable and are given by, for example,

$$p(Y_{x_1} = y_1) = p(u_1) + p(u_2), \quad p(Y_{x_0} = y_1) = p(u_1) + p(u_3). \tag{7}$$

Clearly, Theorem 2 shows that PN, PS and PNS are also identifiable.

## 4  Estimation

When the joint probabilities of potential outcomes are identifiable through the proposed conditions, as seen from the proofs of Theorems (refer to the supplementary material A), the estimation problem of the joint probabilities of potential outcomes reduces to that of singular models and thus they can not be evaluated by standard statistical estimation methods. To solve the problem, we proposes a new statistical method based on the augmented Lagrangian method [3]. In this section, we focus on Case 2 in Section 3.2. For estimation in Case 1, see the supplementary material B.1.

Consider the matrices $\widehat{P}_x$ and $\widehat{Q}_x$ that are derived by replacing $p(z \mid x)$, $p(w \mid x)$, $p(z, w \mid x)$, $p(y \mid x)$, $p(y, z \mid x)$, $p(y, w \mid x)$, and $p(y, z, w \mid x)$ of $P_x$ and $Q_x$ with sample probabilities $\widehat{p}(z \mid x)$, $\widehat{p}(w \mid x)$, $\widehat{p}(z, w \mid x)$, $\widehat{p}(y \mid x)$, $\widehat{p}(y, z \mid x)$, $\widehat{p}(y, w \mid x)$, and $\widehat{p}(y, z, w \mid x)$, respectively ($x \in \{x_1, x_0\}$; $y \in \{y_1, y_0\}$; $z \in \{z_1, \ldots, z_4\}$; $w \in \{w_1, \ldots, w_4\}$). Then, for constant matrices $M_{x_1}$ and $M_{x_0}$ (see equation (A.23) of the supplementary material A.2), since we have

$$SP_x^{-1}Q_x = M_x S, \quad x \in \{x_1, x_0\} \tag{8}$$

from equation (A.26) of the supplementary material A.2, we estimate the matrix $S$ (see equation (A.24) of the supplementary material A.2) as the solution $\Theta = (\theta_{ij})_{1 \leq i, j \leq 4}$ of the optimization problem

$$\underset{\Theta}{\text{minimize}} \; \frac{1}{2} \|\Theta \widehat{P}_{x_1}^{-1} \widehat{Q}_{x_1} - M_{x_1} \Theta\|_F^2 + \frac{1}{2} \|\Theta \widehat{P}_{x_0}^{-1} \widehat{Q}_{x_0} - M_{x_0} \Theta\|_F^2 \tag{9}$$

$$\text{subject to } 0 \leq \boldsymbol{e}_j^\top (\widehat{P}_x \Theta^{-1})^\top \boldsymbol{e}_1 \leq 1, \; j = 1, \ldots, 4; \; \mathbf{1}^\top (\widehat{P}_x \Theta^{-1})^\top \boldsymbol{e}_1 = 1, \; x \in \{x_1, x_0\}, \tag{10}$$

Table 1: Conditional probability tables in simulation.

| | (a) $p(Z \mid U)$ | | | | (b) $p(W \mid U)$ | | | | (c) $p(Y = 1 \mid X, U)$ | |
|---|---|---|---|---|---|---|---|---|---|---|
| | $Z = 1$ | $Z = 2$ | $Z = 3$ | $Z = 4$ | $W = 1$ | $W = 2$ | $W = 3$ | $W = 4$ | $X = 1$ | $X = 0$ |
| $U = 1$ | 7/10 | 1/10 | 1/10 | 1/10 | 7/10 | 1/10 | 1/10 | 1/10 | 1 | 1 |
| $U = 2$ | 1/10 | 7/10 | 1/10 | 1/10 | 1/10 | 7/10 | 1/10 | 1/10 | 1 | 0 |
| $U = 3$ | 1/10 | 1/10 | 7/10 | 1/10 | 1/10 | 1/10 | 7/10 | 1/10 | 0 | 1 |
| $U = 4$ | 1/10 | 1/10 | 1/10 | 7/10 | 1/10 | 1/10 | 1/10 | 7/10 | 0 | 0 |

| | (d) $p(X = 1 \mid W, U)$ | | | | (e) $p(U)$ |
|---|---|---|---|---|---|
| | $W = 1$ | $W = 2$ | $W = 3$ | $W = 4$ | |
| $U = 1$ | 21/46 | 18/43 | 18/43 | 18/43 | 1/4 |
| $U = 2$ | 9/34 | 21/71 | 9/34 | 9/34 | 1/4 |
| $U = 3$ | 9/34 | 9/34 | 21/71 | 9/34 | 1/4 |
| $U = 4$ | 9/34 | 9/34 | 9/34 | 21/71 | 1/4 |

$$\theta_{i1} = 1, \quad 0 \le \theta_{ij} \le 1, \quad \sum_{k=2}^{4} \theta_{ik} \le 1, \quad i = 1, \ldots, 4; \quad j = 2, 3, 4, \tag{11}$$

where $\boldsymbol{e_1} = (1, 0, 0, 0)^\top$, $\boldsymbol{e_2} = (0, 1, 0, 0)^\top$, $\boldsymbol{e_3} = (0, 0, 1, 0)^\top$, $\boldsymbol{e_4} = (0, 0, 0, 1)^\top$, and $\boldsymbol{1} = (1, 1, 1, 1)^\top$. In addition, $\| \cdot \|_F$ denotes the Frobenius norm of a matrix. Equation (10) is the constraint such that the first row of $\widehat{P}_x \Theta^{-1}$ comes to the valid estimators of $(p(u_1|x), \ldots, p(u_4|x))$.

To solve the optimization problem, we use the augmented Lagrangian method [3]. When we obtain the estimator $\widehat{S}$ as the solution of the optimization problem, the estimator of $\boldsymbol{u} = (p(u_1), \ldots, p(u_4))^\top$ is given by the formula

$$\widehat{\boldsymbol{u}} = \widehat{p}(x_1)(\widehat{P}_{x_1} \widehat{S}^{-1})^\top \boldsymbol{e_1} + \widehat{p}(x_0)(\widehat{P}_{x_0} \widehat{S}^{-1})^\top \boldsymbol{e_1}.$$

where $\widehat{p}(x)$ is the sample probability for $p(x)$. For details on the asymptotic normality of the estimators and algorithms, see the supplementary material.

## 5   Numerical Experiment

In this section, we present a numerical experiment to examine properties of the proposed estimation method through the PNS $p(u_2)$ and the causal risk difference $p(u_2) - p(u_3)$. For simplicity, letting $X$, $Y$, $Z$, $W$, and $U$ be discrete variables, we consider the causal diagrams shown in Figure 2, where the joint probabilities of $(X, Y, Z, W, U)$ are given according to Table 1. Under the situation where $(X, Y, Z, W)$ can be observed but $U$ can not, the properties of the proposed estimators $\widehat{p}(u_2)$ and $\widehat{p}(u_2) - \widehat{p}(u_3)$ of $p(u_2)$ and $p(u_2) - p(u_3)$, respectively, are verified in a numerical experiment using the setting with sample sizes $n = 100, 200, 1000$, and $5000$. In this situation, since the $p(u_2)$ and $p(u_2) - p(u_3)$ are 1/4 and zero, respectively, the sample means of $\widehat{p}(u_2)$ and $\widehat{p}(u_2) - \widehat{p}(u_3)$ are expected to be close to 1/4 and zero, respectively. Table 2 and Figure 3 show the basic statistics and the box plots of $\widehat{p}(u_2)$ and $\widehat{p}(u_2) - \widehat{p}(u_3)$ for 1000 replications with the given sample size $n$, respectively. The horizontal lines in Figure 3 show the true values of $p(u_2)$ and $p(u_2) - p(u_3)$.

From Table 2, the sample means of $\widehat{p}(u_2)$ and $\widehat{p}(u_2) - \widehat{p}(u_3)$ are close to the true values and the sample standard deviations are smaller as the sample size is larger. Thus, it seems that the proposed estimation method provides the consistent estimators of $p(u_2)$ and $p(u_2) - p(u_3)$. In addition, from Figure 3, the interquantile ranges for $\widehat{p}(u_2)$ and $\widehat{p}(u_2) - \widehat{p}(u_3)$ are narrower and still include the true values even if the sample size is large. In contrast, it seems that $\widehat{p}(u_2) - \widehat{p}(u_3)$ is symmetrically distributed and thus the asymptotic normality holds, but $\widehat{p}(u_2)$ may not. This is because the true value of $p(u_2)$ is relatively close to zero and Figure 3 is given as the zero truncated distribution of $\widehat{p}(u_2)$ for the finite sample size. Here, note that there are many outliers in each sample size. The outliers would occur when it is difficult to judge that Condition 6 holds from observed data. For further discussion on the simulation experiments, see the supplementary material.

Table 2: Basic statistics of estimates based on the proposed method.

| | (a) $\widehat{p}(u_2)$ | | | | (b) $\widehat{p}(u_2) - \widehat{p}(u_3)$ | | | |
|---|---|---|---|---|---|---|---|---|
| | $n = 100$ | $n = 200$ | $n = 1000$ | $n = 5000$ | $n = 100$ | $n = 200$ | $n = 1000$ | $n = 5000$ |
| Minimum | 0.003 | 0.001 | 0.008 | 0.002 | $-0.906$ | $-0.936$ | $-0.957$ | $-0.864$ |
| 1st Quantile | 0.176 | 0.210 | 0.234 | 0.241 | $-0.107$ | $-0.076$ | $-0.043$ | $-0.033$ |
| Median | 0.242 | 0.253 | 0.258 | 0.254 | $-0.015$ | $-0.006$ | $-0.005$ | $-0.005$ |
| Mean | 0.251 | 0.259 | 0.264 | 0.255 | $-0.026$ | $-0.017$ | $-0.013$ | $-0.025$ |
| 3rd | 0.305 | 0.294 | 0.281 | 0.271 | 0.083 | 0.061 | 0.034 | 0.016 |
| Maximum | 0.890 | 0.915 | 0.874 | 0.726 | 0.833 | 0.893 | 0.867 | 0.527 |
| s.e. | 0.126 | 0.103 | 0.082 | 0.072 | 0.228 | 0.189 | 0.139 | 0.129 |

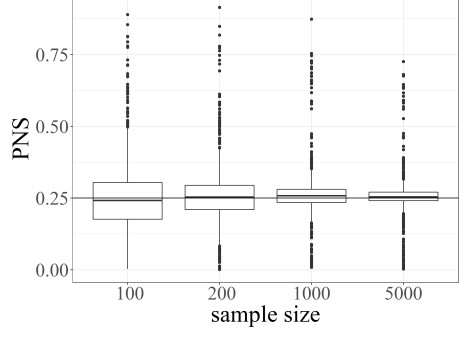

(a) Boxplots for PNS

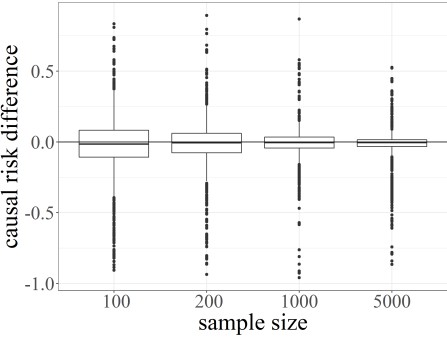

(b) Boxplots for causal risk difference

Figure 3: Boxplots of estimates based on the proposed method

## 6 Discussion

Joint probabilities of potential outcomes often appears in unit selection problems [20], the impact evaluation problem of social programs [12], the non-compliance problem of treatment effects [1, 2], the principal stratification [9], the identification problems of natural direct and indirect effects [25] and the prevented and preventable proportions [36], and the explainability problem of artificial intelligence [10, 22, 35]. In addition, if the joint probabilities of potential outcomes are identifiable, then the causal risks are also identifiable, but not vise versa. Therefore, the identification and estimation problems of these probabilities have been an important topic in causal inference. To solve the problems, the paper proposed (i) two types of novel identification conditions using covariate information and (ii) a new statistical estimation method based on the augmented Lagrangian method [3]. To the best of our knowledge, there has been previously much less discussion how useful covariate information is to identify these probabilities in nonparametric systems when existing identification conditions do not hold. In addition, the estimation problem of singular models often appears in the context of causal inference, but it seems that the solution has not been given in the literature. Thus, the results of this paper extend the range of solvable identification and estimation problems of causal inference under the nonparametric causal models.

Finally, we assumed that both an exposure variable and an outcome variable is dichotomous. However, it is straightforward to extend our results from the case of dichotomous observed variables to the case of multivalued observed variables under certain assumptions. For example, multivalued or continuous outcome can be accommodated in the model using the event $Y < y$ as an outcome variable. Then, for given $y$, the problem of the paper can be the identification and estimation of $p(Y_x > y, Y_{x'} \le y), \ldots, p(Y_x > y, Y_{x'} > y)$ and thus the result of this paper are applicable to the problems. For the related discussion, refer to Galhotra et al. [10] and Kada et al. [14]. In addition, when the exposure variable is continuous, according to Balke and Pearl [2], it is reasonable to assume that there exists an exposure interval around each $x$, within which a subject's outcome is homogeneous. Under this assumption, it is possible to apply our results. However, in such cases, it may be difficult to obtain reliable statistics of the recovered probabilities due to data sparseness. It is left for future work.

## Acknowledgments and Disclosure of Funding

We would like to acknowledge the helpful comments of the four anonymous reviewers. This research was partially funded by JFE Engineering Corporation and Japan Society for the Promotion of Science (JSPS), Grant Number 19K11856 and 21H03504.

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
