# Supplemental Material of "Identification and Estimation of Joint Probabilities of Potential Outcomes in Observational Studies with Covariate Information"

**Ryusei Shingaki**    **Manabu Kuroki**
Graduate School of Engineering Science, Yokohama National University
shingaki-ryusei-kw@ynu.jp    kuroki-manabu-zm@ynu.ac.jp

## A    Proofs of theorems

### A.1    Proof of Theorem 1

From Conditions 1 and 2 in Theorem 1, by the consistency property, we have

$$p(x_1, y_1, z) = \sum_{i=1,2} p(x_1|u_i)p(z|u_i)p(u_i), \quad p(x_0, y_1, z) = \sum_{i=1,3} p(x_0|u_i)p(z|u_i)p(u_i), \quad \text{(A.1)}$$

$$p(x_1, y_0, z) = \sum_{i=3,4} p(x_1|u_i)p(z|u_i)p(u_i), \quad p(x_0, y_0, z) = \sum_{i=2,4} p(x_0|u_i)p(z|u_i)p(u_i), \quad \text{(A.2)}$$

$$p(x_1, y_1) = \sum_{i=1,2} p(x_1|u_i)p(u_i), \quad p(x_0, y_1) = \sum_{i=1,3} p(x_0|u_i)p(u_i), \quad \text{(A.3)}$$

$$p(x_1, y_0) = \sum_{i=3,4} p(x_1|u_i)p(u_i), \quad p(x_0, y_0) = \sum_{i=2,4} p(x_0|u_i)p(u_i), \quad \text{(A.4)}$$

$$p(y_{x_1}, z) = p(y_{x_1}|z)p(z) = \sum_{i=1,2} p(z|u_i)p(u_i), \quad p(y_{x_0}, z) = p(y_{x_0}|z)p(z) = \sum_{i=1,3} p(z|u_i)p(u_i), \quad \text{(A.5)}$$

$$p(z) = \sum_{i=1}^{4} p(z|u_i)p(u_i), \quad p(y_{x_1}) = \sum_{z} p(y_{x_1}, z), \quad p(y_{x_0}) = \sum_{z} p(y_{x_0}, z) \quad \text{(A.6)}$$

for $z \in \{z_1, \ldots, z_4\}$. Thus, letting

$$P = \begin{pmatrix} 1 & p(z_1) & p(z_2) & p(z_3) \\ p(x_1, y_1) & p(x_1, y_1, z_1) & p(x_1, y_1, z_2) & p(x_1, y_1, z_3) \\ p(x_1, y_0) & p(x_1, y_0, z_1) & p(x_1, y_0, z_2) & p(x_1, y_0, z_3) \\ p(x_0, y_1) & p(x_0, y_1, z_1) & p(x_0, y_1, z_2) & p(x_0, y_1, z_3) \end{pmatrix}, \quad \text{(A.7)}$$

$$Q = \begin{pmatrix} 1 & p(z_1) & p(z_2) & p(z_3) \\ p(y_{x_1}) & p(y_{x_1}, z_1) & p(y_{x_1}, z_2) & p(y_{x_1}, z_3) \\ p(y_{x_0}) & p(y_{x_0}, z_1) & p(y_{x_0}, z_2) & p(y_{x_0}, z_3) \end{pmatrix}, \quad \text{(A.8)}$$

$$R = \begin{pmatrix} 1 & p(x_1|u_1) & 0 & p(x_0|u_1) \\ 1 & p(x_1|u_2) & 0 & 0 \\ 1 & 0 & p(x_1|u_3) & p(x_0|u_3) \\ 1 & 0 & p(x_1|u_4) & 0 \end{pmatrix}, \quad S = \begin{pmatrix} 1 & p(z_1|u_1) & p(z_2|u_1) & p(z_3|u_1) \\ 1 & p(z_1|u_2) & p(z_2|u_2) & p(z_3|u_2) \\ 1 & p(z_1|u_3) & p(z_2|u_3) & p(z_3|u_3) \\ 1 & p(z_1|u_4) & p(z_2|u_4) & p(z_3|u_4) \end{pmatrix}, \quad \text{(A.9)}$$

35th Conference on Neural Information Processing Systems (NeurIPS 2021).

$$M = \begin{pmatrix} 1 & 1 & 1 & 1 \\ 1 & 1 & 0 & 0 \\ 1 & 0 & 1 & 0 \end{pmatrix}, \quad \Delta = \begin{pmatrix} p(u_1) & 0 & 0 & 0 \\ 0 & p(u_2) & 0 & 0 \\ 0 & 0 & p(u_3) & 0 \\ 0 & 0 & 0 & p(u_4) \end{pmatrix}, \tag{A.10}$$

we derive

$$P = R^\top \Delta S, \quad Q = M\Delta S, \tag{A.11}$$

where the notation "$\top$" stands for a transposed vector/matrix. Since $P$ is invertible from Condition 3 in Theorem 1, from equation (A.11), $R$ is given as the solution of the simultaneous linear equation

$$QP^{-1}R^\top = M. \tag{A.12}$$

Letting $a_i$ and $m_i$ be the $i$-th column vector of the $3 \times 4$ matrices $QP^{-1}$ and $M$, respectively $(i = 1, \ldots, 4)$, that is, $QP^{-1} = (a_1; a_2; a_3; a_4)$ and $M = (m_1; m_2; m_3; m_4)$. From equation (A.12), since we have

$$QP^{-1}R^\top = (a_1; a_2; a_3; a_4)R^\top = M = (m_1; m_2; m_3; m_4),$$

or, equivalently,

$$a_1 + p(x_1|u_1)a_2 + \{1 - p(x_1|u_1)\} a_4 = m_1, \quad a_1 + p(x_1|u_2)a_2 = m_2,$$
$$a_1 + p(x_1|u_3)a_3 + \{1 - p(x_1|u_3)\} a_4 = m_3, \quad a_1 + p(x_1|u_4)a_3 = m_4,$$

we derive

$$p(x_1|u_1)(a_2 - a_4) = m_1 - a_1 - a_4, \quad p(x_1|u_2)a_2 = m_2 - a_1,$$
$$p(x_1|u_3)(a_3 - a_4) = m_3 - a_1 - a_4, \quad p(x_1|u_4)a_3 = m_4 - a_1.$$

It shows that $p(x_1|u_j)$ and $p(x_0|u_j) = 1 - p(x_1|u_j)$ are identifiable since $a_j$ and $m_j$ are identifiable $(j = 1, \ldots, 4)$ under Condition 3 in Theorem 1. In addition, note that we have $(R^\top)^{-1}P = \Delta S$ from equation (A.11), and the first column of $\Delta S$ is given as $(p(u_1), \ldots, p(u_4))$. Thus, a comparison between the first column of $(R^\top)^{-1}P$ and $\Delta S$ shows that $p(u_1), \ldots, p(u_4)$ are identifiable since both $P$ and $R$ are identifiable. Thus, PN, PS, and PNS are identifiable since

$$p(x_1, u_i) = p(x_1|u_i)p(u_i), \quad p(x_0, u_i) = p(x_0|u_i)p(u_i)$$

are identifiable from $R$, $\Delta$ and $P$ for $i = 1, \ldots, 4$. $\qquad\square$

## A.2  Proof of Theorem 2

From Conditions 4 and 5 of Theorem 2, by the consistency property, we have

$$p(y_1, z, w|x_1) = \sum_{i=1,2} p(w|x_1, u_i)p(z|u_i)p(u_i|x_1), \tag{A.13}$$

$$p(y_1, z, w|x_0) = \sum_{i=1,3} p(w|x_0, u_i)p(z|u_i)p(u_i|x_0), \tag{A.14}$$

$$p(y_0, z, w|x_1) = \sum_{i=3,4} p(w|x_1, u_i)p(z|u_i)p(u_i|x_1), \tag{A.15}$$

$$p(y_0, z, w|x_0) = \sum_{i=2,4} p(w|x_0, u_i)p(z|u_i)p(u_i|x_0), \tag{A.16}$$

$$p(z, w|x_1) = \sum_{i=1}^{4} p(w|x_1, u_i)p(z|u_i)p(u_i|x_1), \quad p(z, w|x_0) = \sum_{i=1}^{4} p(w|x_0, u_i)p(z|u_i)p(u_i|x_0), \tag{A.17}$$

$$p(y_1|x_1) = \sum_{i=1,2} p(u_i|x_1), \quad p(y_1|x_0) = \sum_{i=1,3} p(u_i|x_0) \tag{A.18}$$

for $z \in \{z_1, \ldots, z_4\}$ and $w \in \{w_1, \ldots, w_4\}$. Then, for $x \in \{x_1, x_0\}$, letting

$$P_x = \begin{pmatrix} 1 & p(z_1|x) & p(z_2|x) & p(z_3|x) \\ p(w_1|x) & p(z_1, w_1|x) & p(z_2, w_1|x) & p(z_3, w_1|x) \\ p(w_2|x) & p(z_1, w_2|x) & p(z_2, w_2|x) & p(z_3, w_2|x) \\ p(w_3|x) & p(z_1, w_3|x) & p(z_2, w_3|x) & p(z_3, w_3|x) \end{pmatrix}, \tag{A.19}$$

$$Q_x = \begin{pmatrix} p(y_1|x) & p(y_1,z_1|x) & p(y_1,z_2|x) & p(y_1,z_3|x) \\ p(y_1,w_1|x) & p(y_1,z_1,w_1|x) & p(y_1,z_2,w_1|x) & p(y_1,z_3,w_1|x) \\ p(y_1,w_2|x) & p(y_1,z_1,w_2|x) & p(y_1,z_2,w_2|x) & p(y_1,z_3,w_2|x) \\ p(y_1,w_3|x) & p(y_1,z_1,w_3|x) & p(y_1,z_2,w_3|x) & p(y_1,z_3,w_3|x) \end{pmatrix}, \tag{A.20}$$

$$R_x = \begin{pmatrix} 1 & p(w_1|x,u_1) & p(w_2|x,u_1) & p(w_3|x,u_1) \\ 1 & p(w_1|x,u_2) & p(w_2|x,u_2) & p(w_3|x,u_2) \\ 1 & p(w_1|x,u_3) & p(w_2|x,u_3) & p(w_3|x,u_3) \\ 1 & p(w_1|x,u_4) & p(w_2|x,u_4) & p(w_3|x,u_4) \end{pmatrix}, \tag{A.21}$$

$$\Delta_x = \begin{pmatrix} p(u_1|x) & 0 & 0 & 0 \\ 0 & p(u_2|x) & 0 & 0 \\ 0 & 0 & p(u_3|x) & 0 \\ 0 & 0 & 0 & p(u_4|x) \end{pmatrix}, \tag{A.22}$$

$$M_{x_1} = \begin{pmatrix} 1 & 0 & 0 & 0 \\ 0 & 1 & 0 & 0 \\ 0 & 0 & 0 & 0 \\ 0 & 0 & 0 & 0 \end{pmatrix}, \quad M_{x_0} = \begin{pmatrix} 1 & 0 & 0 & 0 \\ 0 & 0 & 0 & 0 \\ 0 & 0 & 1 & 0 \\ 0 & 0 & 0 & 0 \end{pmatrix}, \tag{A.23}$$

$$S = \begin{pmatrix} 1 & p(z_1|u_1) & p(z_2|u_1) & p(z_3|u_1) \\ 1 & p(z_1|u_2) & p(z_2|u_2) & p(z_3|u_2) \\ 1 & p(z_1|u_3) & p(z_2|u_3) & p(z_3|u_3) \\ 1 & p(z_1|u_4) & p(z_2|u_4) & p(z_3|u_4) \end{pmatrix}, \tag{A.24}$$

we derive

$$P_x = R_x^\top \Delta_x S, \quad Q_x = R_x^\top \Delta_x M_x S \quad \text{for } x \in \{x_1, x_0\}. \tag{A.25}$$

Here, since both $P_{x_1}$ and $P_{x_0}$ are invertible from Condition 6 in Theorem 2, we obtain

$$P_x^{-1} Q_x = S^{-1} M_x S \quad \text{for } x \in \{x_1, x_0\}, \tag{A.26}$$

whose eigenvalues are 0 and 1. Then, letting $E_{x_1}$ and $E_{x_0}$ be $4 \times 4$ matrices

$$E_{x_1} = \begin{pmatrix} e_{11} & e_{12} & 0 & 0 \\ e_{21} & e_{22} & 0 & 0 \\ 0 & 0 & e_{33} & e_{34} \\ 0 & 0 & e_{43} & e_{44} \end{pmatrix}, \quad E_{x_0} = \begin{pmatrix} e'_{11} & 0 & e'_{13} & 0 \\ 0 & e'_{22} & 0 & e'_{24} \\ e'_{31} & 0 & e'_{33} & 0 \\ 0 & e'_{42} & 0 & e'_{44} \end{pmatrix},$$

the matrices of eigenvectors of $P_{x_1}^{-1} Q_{x_1}$ and $P_{x_0}^{-1} Q_{x_0}$ can be written by $A_{x_1} = S^{-1} E_{x_1}$ and $A_{x_0} = S^{-1} E_{x_0}$, respectively. Thus, we have

$$A_{x_1} E_{x_1}^{-1} = S^{-1} = A_{x_0} E_{x_0}^{-1}.$$

Letting $A_{x_1}^{-1} = (a^{ij})$ and $A_{x_1}^{-1} A_{x_0} = (b^{ij})$, from $E_{x_1} A_{x_1}^{-1} A_{x_0} = E_{x_0}$ and the first column of $E_{x_1} A_{x_1}^{-1} = S$, we have

$$\begin{pmatrix} e_{11} & e_{12} \\ e_{21} & e_{22} \end{pmatrix} \begin{pmatrix} a^{11} & b^{11} & b^{12} \\ a^{21} & b^{21} & b^{22} \end{pmatrix} = \begin{pmatrix} 1 & e'_{11} & 0 \\ 1 & 0 & e'_{22} \end{pmatrix},$$
$$\begin{pmatrix} e_{33} & e_{34} \\ e_{43} & e_{44} \end{pmatrix} \begin{pmatrix} a^{31} & b^{31} & b^{32} \\ a^{41} & b^{41} & b^{42} \end{pmatrix} = \begin{pmatrix} 1 & e'_{31} & 0 \\ 1 & 0 & e'_{42} \end{pmatrix} \tag{A.27}$$

From equation (A.27), $e_{21}$, $e_{22}$, $e_{43}$, and $e_{44}$ are identifiable by noting the second row of each of the following matrices:

$$\begin{pmatrix} e_{11} & e_{12} \\ e_{21} & e_{22} \end{pmatrix} = \begin{pmatrix} 1 & e'_{11} \\ 1 & 0 \end{pmatrix} \begin{pmatrix} a^{11} & b^{11} \\ a^{21} & b^{21} \end{pmatrix}^{-1},$$
$$\begin{pmatrix} e_{33} & e_{34} \\ e_{43} & e_{44} \end{pmatrix} = \begin{pmatrix} 1 & e'_{31} \\ 1 & 0 \end{pmatrix} \begin{pmatrix} a^{31} & b^{31} \\ a^{41} & b^{41} \end{pmatrix}^{-1}. \tag{A.28}$$

Similarly, $e_{11}$, $e_{12}$, $e_{33}$, and $e_{34}$ are identifiable by noting the first row of each of the following matrices:

$$\begin{pmatrix} e_{11} & e_{12} \\ e_{21} & e_{22} \end{pmatrix} = \begin{pmatrix} 1 & 0 \\ 1 & e'_{22} \end{pmatrix} \begin{pmatrix} a^{11} & b^{12} \\ a^{21} & b^{22} \end{pmatrix}^{-1},$$
$$\begin{pmatrix} e_{33} & e_{34} \\ e_{43} & e_{44} \end{pmatrix} = \begin{pmatrix} 1 & 0 \\ 1 & e'_{42} \end{pmatrix} \begin{pmatrix} a^{31} & b^{32} \\ a^{41} & b^{42} \end{pmatrix}^{-1}. \tag{A.29}$$

Thus, $E_{x_1}$ is identifiable and thus $E_{x_0}$ and $S$ are identifiable from $E_{x_1} A_{x_1}^{-1} A_{x_0} = E_{x_0}$ and $E_{x_1} A_{x_1}^{-1} = S$, respectively.

Then, a comparison between the first row of $P_{x_1} S^{-1}$ and $R_{x_1}^\top \Delta_{x_1}$ shows that $p(u_1|x_1), \ldots, p(u_4|x_1)$ are identifiable since both $P_{x_1}$ and $S$ are identifiable and the first row of $R_{x_1}^\top \Delta_{x_1}$ are given as $(p(u_1|x_1), \ldots, p(u_4|x_1))$. Similarly, $p(u_1|x_0), \ldots, p(u_4|x_0)$ are identifiable as ascertained through a comparison between the first row of $P_{x_0} S^{-1}$ and $R_{x_0}^\top \Delta_{x_0}$. Since

$$p(u_i) = p(u_i|x_1)p(x_1) + p(u_i|x_0)p(x_0),$$

the PN, PS, and PNS are identifiable for $i = 1, \ldots, 4$. □

## B Estimation

### B.1 Estimation in Case 1

Let $\{(X_i, Y_i, Z_i)\}_{i=1}^n$ be a sample from the data generating process in Figure 1. In addition, we observe $W$ where $\{Z, W\}$ satisfies the back-door criterion to relative to $(X, Y)$ and let denote the sample as $\{W_i\}_{i=1}^n$. Let denote the maximum likelihood estimators of $p(z|x)$, $p(y|x)$, $p(w|z)$, $p(y, z|x)$, and $p(y|x, z, w)$ as $\widehat{p}(z|x)$, $\widehat{p}(y|x)$, $\widehat{p}(w|z)$, $\widehat{p}(y, z|x)$, and $\widehat{p}(y|x, z, w)$ for $x \in \{x_1, x_0\}$, $y \in \{y_1, y_0\}$, and $z \in \{z_1, \ldots, z_4\}$, respectively. Then, since $p(Y_x = y|z)$ is identifiable and is given by

$$\widehat{p}(Y_x = y|z) = \sum_w \widehat{p}(y|x, z, w)\widehat{p}(w|z), \tag{B.1}$$

$p(y_x)$, $p(y_x, z_1)$, $p(y_x, z_2)$, and $p(y_x, z_3)$ are also identifiable. We denote the estimators as $\widehat{p}(y_x)$, $\widehat{p}(y_x, z_1)$, $\widehat{p}(y_x, z_2)$, and $\widehat{p}(y_x, z_3)$. Let the plug-in estimators of $P$ and $Q$ denote as

$$\widehat{P} = \begin{pmatrix} 1 & \widehat{p}(z_1) & \widehat{p}(z_2) & \widehat{p}(z_3) \\ \widehat{p}(x_1, y_1) & \widehat{p}(x_1, y_1, z_1) & \widehat{p}(x_1, y_1, z_2) & \widehat{p}(x_1, y_1, z_3) \\ \widehat{p}(x_1, y_0) & \widehat{p}(x_1, y_0, z_1) & \widehat{p}(x_1, y_0, z_2) & \widehat{p}(x_1, y_0, z_3) \\ \widehat{p}(x_0, y_1) & \widehat{p}(x_0, y_1, z_1) & \widehat{p}(x_0, y_1, z_2) & \widehat{p}(x_0, y_1, z_3) \end{pmatrix}, \tag{B.2}$$

$$\widehat{Q} = \begin{pmatrix} 1 & \widehat{p}(z_1) & \widehat{p}(z_2) & \widehat{p}(z_3) \\ \widehat{p}(y_{x_1}) & \widehat{p}(y_{x_1}, z_1) & \widehat{p}(y_{x_1}, z_2) & \widehat{p}(y_{x_1}, z_3) \\ \widehat{p}(y_{x_0}) & \widehat{p}(y_{x_0}, z_1) & \widehat{p}(y_{x_0}, z_2) & \widehat{p}(y_{x_0}, z_3) \end{pmatrix}. \tag{B.3}$$

From the proof of Theorem 1 in the Supplementary Material A.1, given $P$ and $Q$, the identifiable matrix $R$ satisfies

$$QP^{-1}R^\top = M.$$

It means that $R$ is a solution of the following minimization problem

$$\underset{\Theta \in \mathcal{T}}{\text{minimize}} \ \frac{1}{2}\|QP^{-1}\Theta^\top - M\|_F^2 \tag{B.4}$$

$$\text{subject to } 0 \le (\Theta^\top)^{-1} P e_1 \le 1, \quad \mathbf{1}^\top (\Theta^\top)^{-1} P e_1 = 1, \quad \theta_{12} + \theta_{14} = \theta_{33} + \theta_{34} = 1, \tag{B.5}$$

where $e_1 = (1, 0, 0, 0)^\top$, $e_2 = (0, 1, 0, 0)^\top$, $e_3 = (0, 0, 1, 0)^\top$, $e_4 = (0, 0, 0, 1)^\top$, $\mathbf{1} = (1, 1, 1, 1)^\top$ and

$$\mathcal{T} := \{\Theta = (\theta_{ij}) \in GL_4(\mathbb{R}) : \theta_{i1} = 1, \theta_{32} = \theta_{42} = \theta_{13} = \theta_{23} = \theta_{24} = \theta_{44} = 0, \text{ and } 0 \le \Theta \le 1\}.$$

Here, $GL_4(\mathbb{R})$ is the group of invertible $4 \times 4$ matrices with entries in $\mathbb{R}$ and inequalities are understood component-wise. The equation (B.5) is the condition in which the first column of $(\Theta^\top)^{-1}P$ is consistent with $(p(u_1), \ldots, p(u_4))$.

We propose to estimate $R$ as a solution of the following minimization problem by replacing $P$ and $Q$ to $\widehat{P}$ and $\widehat{Q}$, respectively,

$$\underset{\Theta \in \mathcal{T}}{\text{minimize}} \ \frac{1}{2}\|\widehat{Q}\widehat{P}^{-1}\Theta^\top - M\|_F^2 \tag{B.6}$$

$$\text{subject to } 0 \le (\Theta^\top)^{-1} \widehat{P} e_1 \le 1, \quad \mathbf{1}^\top (\Theta^\top)^{-1} \widehat{P} e_1 = 1, \quad \theta_{12} + \theta_{14} = \theta_{33} + \theta_{34} = 1, \tag{B.7}$$

Following Bertsekas [2], let denote the augmented Lagrangians as

$$
\begin{aligned}
&L(\Theta; \widehat{P}, \widehat{Q}, \mu, \mu_1, \mu_2, \boldsymbol{\lambda}) \\
&= \frac{1}{2}\|\widehat{Q}\widehat{P}^{-1}\Theta^\top - M\|_F^2 + \mu\left(\mathbf{1}^\top(\Theta^\top)^{-1}\widehat{P}\boldsymbol{e}_1 - 1\right) + \frac{\rho}{2}\left(\mathbf{1}^\top(\Theta^\top)^{-1}\widehat{P}\boldsymbol{e}_1 - 1\right)^2 \\
&\quad + \mu_1\left(\theta_{12} + \theta_{14} - 1\right) + \frac{\rho}{2}\left(\theta_{12} + \theta_{14} - 1\right)^2 + \mu_2\left(\theta_{33} + \theta_{34} - 1\right) + \frac{\rho}{2}\left(\theta_{33} + \theta_{34} - 1\right)^2 \\
&\quad + \sum_{j=1}^4 f_j(\Theta, \boldsymbol{\lambda}, \rho),
\end{aligned}
\tag{B.8}
$$

where $\mu$, $\mu_1$, $\mu_2$, and $\boldsymbol{\lambda} = (\lambda_1, \ldots, \lambda_4)^\top$ are the Lagrange multipliers and

$$
\begin{aligned}
f_j(\Theta, \boldsymbol{\lambda}, \rho) &= \min_{0 \le \boldsymbol{e}_j^\top(\Theta^\top)^{-1}\widehat{P}\boldsymbol{e}_1 - v_j \le 1}\left(\lambda_j v_j + \frac{\rho}{2}v_j^2\right) \\
&= \begin{cases}
\lambda_j(\boldsymbol{e}_j^\top(\Theta^\top)^{-1}\widehat{P}\boldsymbol{e}_1 - 1) + \frac{\rho}{2}\left|\boldsymbol{e}_j^\top(\Theta^\top)^{-1}\widehat{P}\boldsymbol{e}_1 - 1\right|^2 & \text{if } \lambda_j + \rho(\boldsymbol{e}_j^\top(\Theta^\top)^{-1}\widehat{P}\boldsymbol{e}_1 - 1) > 1, \\
\lambda_j\boldsymbol{e}_j^\top(\Theta^\top)^{-1}\widehat{P}\boldsymbol{e}_1 + \frac{\rho}{2}\left|\boldsymbol{e}_j^\top(\Theta^\top)^{-1}\widehat{P}\boldsymbol{e}_1\right|^2 & \text{if } \lambda_j + \rho\boldsymbol{e}_j^\top(\Theta^\top)^{-1}\widehat{P}\boldsymbol{e}_1 < 1, \\
-\frac{\lambda_j^2}{2\rho} & \text{otherwise.}
\end{cases}
\end{aligned}
\tag{B.9}
$$

The multiplier iterations are given by

$$
\mu^{(t+1)} = \mu^{(t)} + \rho(\mathbf{1}^\top(\Theta^\top)^{-1}\widehat{P}\boldsymbol{e}_1 - 1),
\tag{B.10}
$$

$$
\mu_1^{(t+1)} = \mu_1^{(t)} + \rho(\theta_{12} + \theta_{14} - 1),
\tag{B.11}
$$

$$
\mu_2^{(t+1)} = \mu_2^{(t)} + \rho(\theta_{33} + \theta_{34} - 1),
\tag{B.12}
$$

$$
\lambda_j^{(t+1)} = \begin{cases}
\lambda_j^{(t)} + \rho(\boldsymbol{e}_j^\top(\Theta^\top)^{-1}\widehat{P}\boldsymbol{e}_1 - 1) & \text{if } \lambda_j^{(t)} + \rho(\boldsymbol{e}_j^\top(\Theta^\top)^{-1}\widehat{P}\boldsymbol{e}_1 - 1) > 1, \\
\lambda_j^{(t)} + \rho\boldsymbol{e}_j^\top(\Theta^\top)^{-1}\widehat{P}\boldsymbol{e}_1 & \text{if } \lambda_j^{(t)} + \rho\boldsymbol{e}_j^\top(\Theta^\top)^{-1}\widehat{P}\boldsymbol{e}_1 < 1, \\
0 & \text{otherwise.}
\end{cases}
\tag{B.13}
$$

Then, the estimator of $R$ is given by the solution $\widehat{\Theta}$ of the following estimating equation

$$
\frac{\partial}{\partial \Theta}L(\Theta; \widehat{P}, \widehat{Q}, \mu, \mu_1, \mu_2, \boldsymbol{\lambda}) = 0.
\tag{B.14}
$$

Algorithm 1 is an algorithm that provides the solutions of the optimization problem based on the augmented Lagrangian method and the update rules via gradient descent. Here, $\alpha$ is the fixed step size at the $t$-th iteration, $T$ is the number of iterations, and $\Theta^{(0)}$ is the initial point. Once we obtain the estimator $\widehat{R}$ as the solution of the optimization problem (B.6), the estimator of $\boldsymbol{u} = (p(u_1), \ldots, p(u_4))^\top$ is given by

$$
\widehat{\boldsymbol{u}} = (\widehat{R}^\top)^{-1}\widehat{P}\boldsymbol{e}_1.
$$

For example, since PNS is the second component of $\boldsymbol{u}$, we can estimate PNS as the second component of $\widehat{\boldsymbol{u}}$. Similarly, we can estimate causal risk difference as the difference between the second and third components of $\widehat{\boldsymbol{u}}$.

## B.2 Estimation in Case 2

Let $\{(X_i, Y_i, Z_i, W_i)\}_{i=1}^n$ be a sample from the data generating process in Figure 2. Let denote the maximum likelihood estimators of $p(z|x)$, $p(w|x)$, $p(z, w|x)$, $p(y|x)$, $p(y, z|x)$, $p(y, w|x)$, and $p(y, z, w|x)$ as $\widehat{p}(z|x)$, $\widehat{p}(w|x)$, $\widehat{p}(z, w|x)$, $\widehat{p}(y|x)$, $\widehat{p}(y, z|x)$, $\widehat{p}(y, w|x)$, and $\widehat{p}(y, z, w|x)$ for $x \in \{x_1, x_0\}$, $y \in \{y_1, y_0\}$, $z \in \{z_1, \ldots, z_4\}$, and $w \in \{w_1, \ldots, w_4\}$, respectively. Then, let the plug-in estimators of $P_x$ and $Q_x$ denote as

$$
\widehat{P}_x = \begin{pmatrix}
1 & \widehat{p}(z_1|x) & \widehat{p}(z_2|x) & \widehat{p}(z_3|x) \\
\widehat{p}(w_1|x) & \widehat{p}(z_1, w_1|x) & \widehat{p}(z_2, w_1|x) & \widehat{p}(z_3, w_1|x) \\
\widehat{p}(w_2|x) & \widehat{p}(z_1, w_2|x) & \widehat{p}(z_2, w_2|x) & \widehat{p}(z_3, w_2|x) \\
\widehat{p}(w_3|x) & \widehat{p}(z_1, w_3|x) & \widehat{p}(z_2, w_3|x) & \widehat{p}(z_3, w_3|x)
\end{pmatrix},
\tag{B.15}
$$

---

**Algorithm 1** Estimation of $\boldsymbol{u} = (p(u_1), \ldots, p(u_4))^\top$ in Case 1.

---

**Input:** $\{(X_i, Y_i, Z_i, W_i)\}_{i=1}^n$, $\alpha$, $T$, $\rho$
**Output:** $\widehat{\boldsymbol{u}}$

1: Initialize $\Theta^{(0)}$, $\mu^{(0)} \leftarrow 0$, $\mu_1^{(0)} \leftarrow 0$, $\mu_2^{(0)} \leftarrow 0$, and $\boldsymbol{\lambda}^{(0)} \leftarrow 0$
2: Calculate $\widehat{p}(y_x)$, $\widehat{p}(y_x, z_1)$, $\widehat{p}(y_x, z_2)$, and $\widehat{p}(y_x, z_3)$ using equation (B.1)
3: Calculate $\widehat{P}$ and $\widehat{Q}$ using observational data $\{(X_i, Y_i, Z_i)\}_{i=1}^n$
4: **for** $t = 0, 1, \ldots, T-1$ **do**
5:     $\theta_{12}^{(t)} \leftarrow \max\{\min\{\theta_{12}^{(t)}, 1\}, 0\}$, $\theta_{22}^{(t)} \leftarrow \max\{\min\{\theta_{22}^{(t)}, 1\}, 0\}$, $\theta_{33}^{(t)} \leftarrow \max\{\min\{\theta_{33}^{(t)}, 1\}, 0\}$,
      $\theta_{43}^{(t)} \leftarrow \max\{\min\{\theta_{43}^{(t)}, 1\}, 0\}$, $\theta_{14}^{(t)} \leftarrow \max\{\min\{\theta_{14}^{(t)}, 1\}, 0\}$, $\theta_{34}^{(t)} \leftarrow \max\{\min\{\theta_{34}^{(t)}, 1\}, 0\}$

6:     $\Theta^{(t+1)} \leftarrow \Theta^{(t)} - \alpha \frac{\partial}{\partial \Theta} L(\Theta; \widehat{P}, \widehat{Q}, \mu^{(t)}, \boldsymbol{\lambda}^{(t)})\big|_{\Theta = \Theta^{(t)}}$
7:     $\mu^{(t+1)} \leftarrow \mu^{(t)} + \rho(\mathbf{1}^\top (\Theta^\top)^{-1} \widehat{P} \boldsymbol{e}_1 - 1)$
8:     $\mu_1^{(t+1)} \leftarrow \mu_1^{(t)} + \rho(\widehat{\theta}_{12} + \widehat{\theta}_{14} - 1)$
9:     $\mu_2^{(t+1)} \leftarrow \mu_2^{(t)} + \rho(\widehat{\theta}_{33} + \widehat{\theta}_{34} - 1)$
10:     **for** $j = 1, \ldots, 4$ **do**
11:        **if** $\lambda_j^{(t)} + \rho(\boldsymbol{e}_j^\top (\Theta^{(t+1)\top})^{-1} \widehat{P} \boldsymbol{e}_1 - 1) > 1$ **then**
12:          $\lambda_j^{(t+1)} \leftarrow \lambda_j^{(t)} + \rho(\boldsymbol{e}_j^\top (\Theta^{(t+1)\top})^{-1} \widehat{P} \boldsymbol{e}_1 - 1)$
13:        **else if** $\lambda_j^{(t)} + \rho \boldsymbol{e}_j^\top (\Theta^{(t+1)\top})^{-1} \widehat{P} \boldsymbol{e}_1 < 1$ **then**
14:          $\lambda_{x_1, j}^{(t+1)} \leftarrow \lambda_j^{(t)} + \rho \boldsymbol{e}_j^\top (\Theta^{(t+1)\top})^{-1} \widehat{P} \boldsymbol{e}_1$
15:        **else**
16:          $\lambda_j^{(t+1)} \leftarrow 0$; $\lambda_j^{(t+1)} \leftarrow 0$
17:        **end if**
18:     **end for**
19: **end for**
20: $\widehat{R} \leftarrow \Theta^{(T)}$
21: $\widehat{\boldsymbol{u}} \leftarrow (\widehat{R}^\top)^{-1} \widehat{P} \boldsymbol{e}_1$

---

$$\widehat{Q}_x = \begin{pmatrix} \widehat{p}(y|x) & \widehat{p}(y, z_1|x) & \widehat{p}(y, z_2|x) & \widehat{p}(y, z_3|x) \\ \widehat{p}(y, w_1|x) & \widehat{p}(y, z_1, w_1|x) & \widehat{p}(y, z_2, w_1|x) & \widehat{p}(y, z_3, w_1|x) \\ \widehat{p}(y, w_2|x) & \widehat{p}(y, z_1, w_2|x) & \widehat{p}(y, z_2, w_2|x) & \widehat{p}(y, z_3, w_2|x) \\ \widehat{p}(y, w_3|x) & \widehat{p}(y, z_1, w_3|x) & \widehat{p}(y, z_2, w_3|x) & \widehat{p}(y, z_3, w_3|x) \end{pmatrix} \tag{B.16}$$

for $x \in \{x_1, x_0\}$. From the proof of Theorem 2 in the Supplemental Material A.2, given $P_x$ and $Q_x$ for $x \in \{x_1, x_0\}$, the identifiable matrix $S$ satisfies

$$P_x = R_x^\top \Delta_x S, \quad Q_x = R_x^\top \Delta_x M_x S, \tag{B.17}$$

thus, we have

$$S P_x^{-1} Q_x = M_x S.$$

Because it means that $S$ is a solution of the following minimization problem

$$\underset{\Theta \in \mathcal{T}}{\text{minimize}} \frac{1}{2} \|\Theta P_{x_1}^{-1} Q_{x_1} - M_{x_1} \Theta\|_F^2 + \frac{1}{2} \|\Theta P_{x_0}^{-1} Q_{x_0} - M_{x_0} \Theta\|_F^2 \tag{B.18}$$

$$\text{subject to } 0 \le (P_{x_1} \Theta^{-1})^\top \boldsymbol{e}_1 \le 1, \quad \mathbf{1}^\top (P_{x_1} \Theta^{-1})^\top \boldsymbol{e}_1 = 1, \tag{B.19}$$

$$0 \le (P_{x_0} \Theta^{-1})^\top \boldsymbol{e}_1 \le 1, \quad \mathbf{1}^\top (P_{x_0} \Theta^{-1})^\top \boldsymbol{e}_1 = 1, \tag{B.20}$$

where

$$\mathcal{T} := \left\{ \Theta = (\theta_{ij}) \in GL_4(\mathbb{R}) \colon \theta_{i1} = 1, \sum_{j=2}^4 \theta_{ij} \le 1 \text{ for } i = 1, \ldots, 4 \text{ and } 0 \le \Theta \le 1 \right\},$$

we propose to estimate $S$ as a solution of the following minimization problem by replacing $P_x$ and $Q_x$ to $\widehat{P}_x$ and $\widehat{Q}_x$, respectively,

$$\underset{\Theta \in \mathcal{T}}{\text{minimize}} \frac{1}{2} \|\Theta \widehat{P}_{x_1}^{-1} \widehat{Q}_{x_1} - M_{x_1} \Theta\|_F^2 + \frac{1}{2} \|\Theta \widehat{P}_{x_0}^{-1} \widehat{Q}_{x_0} - M_{x_0} \Theta\|_F^2 \tag{B.21}$$

$$\text{subject to } 0 \leq (\widehat{P}_{x_1}\Theta^{-1})^\top \boldsymbol{e}_1 \leq 1, \quad \mathbf{1}^\top (\widehat{P}_{x_1}\Theta^{-1})^\top \boldsymbol{e}_1 = 1, \tag{B.22}$$

$$0 \leq (\widehat{P}_{x_0}\Theta^{-1})^\top \boldsymbol{e}_1 \leq 1, \quad \mathbf{1}^\top (\widehat{P}_{x_0}\Theta^{-1})^\top \boldsymbol{e}_1 = 1. \tag{B.23}$$

Here, $GL_4(\mathbb{R})$ is the group of invertible $4 \times 4$ matrices with entries in $\mathbb{R}$ and inequalities are understood component-wise. The equations (B.19) and (B.20) are the conditions in which the first row of $P_x\Theta^{-1}$ is consistent with $(p(u_1|x), \ldots, p(u_4|x))$ for $x \in \{x_0, x_1\}$.

Following Bertsekas [2], let denote the augmented Lagrangians as

$$L(\Theta; \widehat{P}_{x_1}, \widehat{P}_{x_0}, \widehat{Q}_{x_1}, \widehat{Q}_{x_0}, \mu, \boldsymbol{\lambda}_{x_1}, \boldsymbol{\lambda}_{x_0})$$

$$= \sum_{x \in \{x_1, x_0\}} \frac{1}{2} \|\Theta \widehat{P}_x^{-1} \widehat{Q}_x - M_x \Theta\|_F^2 + \sum_{x \in \{x_1, x_0\}} \mu \left(\mathbf{1}^\top (\widehat{P}_x\Theta^{-1})^\top \boldsymbol{e}_1 - 1\right)$$

$$+ \sum_{x \in \{x_1, x_0\}} \frac{\rho}{2} \left(\mathbf{1}^\top (\widehat{P}_x\Theta^{-1})^\top \boldsymbol{e}_1 - 1\right)^2 + \sum_{x \in \{x_1, x_0\}} \sum_{j=1}^{4} f_{x,j}(\Theta, \boldsymbol{\lambda}_x, \rho), \tag{B.24}$$

where $\mu$ and $\boldsymbol{\lambda}_x = (\lambda_{x,1}, \ldots, \lambda_{x,4})^\top$ are the Lagrange multipliers and

$$f_{x,j}(\Theta, \boldsymbol{\lambda}_x, \rho) = \min_{0 \leq \boldsymbol{e}_j^\top (\widehat{P}_x\Theta^{-1})^\top \boldsymbol{e}_1 - v_j \leq 1} \left(\lambda_{x,j} v_j + \frac{\rho}{2} v_j^2\right)$$

$$= \begin{cases} \lambda_{x,j}(\boldsymbol{e}_j^\top (\widehat{P}_x\Theta^{-1})^\top \boldsymbol{e}_1 - 1) + \frac{\rho}{2}\left|\boldsymbol{e}_j^\top (\widehat{P}_x\Theta^{-1})^\top \boldsymbol{e}_1 - 1\right|^2 & \text{if } \lambda_{x,j} + \rho(\boldsymbol{e}_j^\top (\widehat{P}_x\Theta^{-1})^\top \boldsymbol{e}_1 - 1) > 1, \\ \lambda_{x,j}(\boldsymbol{e}_j^\top (\widehat{P}_x\Theta^{-1})^\top \boldsymbol{e}_1) + \frac{\rho}{2}\left|\boldsymbol{e}_j^\top (\widehat{P}_x\Theta^{-1})^\top \boldsymbol{e}_1\right|^2 & \text{if } \lambda_{x,j} + \rho(\boldsymbol{e}_j^\top (\widehat{P}_x\Theta^{-1})^\top \boldsymbol{e}_1) < 1, \\ -\frac{\lambda_{x,j}^2}{2\rho} & \text{otherwise} \end{cases} \tag{B.25}$$

for $x \in \{x_1, x_0\}$. The multiplier iterations are given by

$$\mu^{(t+1)} = \mu^{(t)} + \rho \left(\mathbf{1}^\top (\widehat{P}_x\Theta^{-1})^\top \boldsymbol{e}_1 - 1\right), \tag{B.26}$$

$$\lambda_{x,j}^{(t+1)} = \begin{cases} \lambda_{x,j}^{(t)} + \rho(\boldsymbol{e}_j^\top (\widehat{P}_x\Theta^{-1})^\top \boldsymbol{e}_1 - 1) & \text{if } \lambda_{x,j}^{(t)} + \rho(\boldsymbol{e}_j^\top (\widehat{P}_x\Theta^{-1})^\top \boldsymbol{e}_1 - 1) > 1, \\ \lambda_{x,j}^{(t)} + \rho \boldsymbol{e}_j^\top (\widehat{P}_x\Theta^{-1})^\top \boldsymbol{e}_1 & \text{if } \lambda_{x,j}^{(t)} + \rho \boldsymbol{e}_j^\top (\widehat{P}_x\Theta^{-1})^\top \boldsymbol{e}_1 < 1, \\ 0 & \text{otherwise.} \end{cases} \tag{B.27}$$

Then, the candidate of the estimator of $S$ is given by the solution $\widehat{\Theta}$ of the following estimating equation

$$\frac{\partial}{\partial \Theta} L(\Theta; \widehat{P}_{x_1}, \widehat{P}_{x_0}, \widehat{Q}_{x_1}, \widehat{Q}_{x_0}, \mu, \boldsymbol{\lambda}_{x_1}, \boldsymbol{\lambda}_{x_0}) = 0. \tag{B.28}$$

Algorithm 2 is an algorithm that provides the solutions of the optimization problem based on the augmented Lagrangian method and the update rules via gradient descent. Here, $\alpha$ is the fixed step size at the $t$-th iteration, $T$ is the number of iterations, and $\Theta^{(0)}$ is the initial point. As we can see immediately, for the zero $\widehat{\Theta}$ of the estimating equation (B.28), the any row permutated matrix $\Pi\widehat{\Theta}$ is also the solution of the same estimating equation, where $\Pi$ is the permutation matrix. Therefore, we find the row permutated matrix $\Pi\Theta$, which achieve the smallest losses and adopt the matrix as the estimator of $S$. Once we obtain the estimator $\widehat{\Theta}$ as the solution of the optimization problem (B.21), the estimator of $\boldsymbol{u} = (p(u_1), \ldots, p(u_4))^\top$ is given by

$$\widehat{\boldsymbol{u}} = \left(\frac{1}{n}\sum_{i=1}^{n} \mathbf{1}\{X_i = x_1\}\right)(\widehat{P}_{x_1}\widehat{\Theta}^{-1})^\top \boldsymbol{e}_1 + \left(\frac{1}{n}\sum_{i=1}^{n} \mathbf{1}\{X_i = x_0\}\right)(\widehat{P}_{x_0}\widehat{\Theta}^{-1})^\top \boldsymbol{e}_1.$$

## B.3 Asymptotic normality

Following Yuan and Jennrich [5], we show the asymptotic normality of the estimators from Algorithm 2.

**Algorithm 2** Estimation of $\boldsymbol{u} = (p(u_1), \ldots, p(u_4))^\top$ in Case 2.

---

**Input:** $\{(X_i, Y_i, Z_i, W_i)\}_{i=1}^n$, $\alpha$, $T$, $\rho$
**Output:** $\widehat{\boldsymbol{u}}$
1: Initialize $\Theta^{(0)}$, $\mu^{(0)} \leftarrow 0$, $\boldsymbol{\lambda}_{x_1}^{(0)} \leftarrow 0$ and $\boldsymbol{\lambda}_{x_0}^{(0)} \leftarrow 0$
2: Calculate $\widehat{P}_{x_1}$, $\widehat{P}_{x_0}$, $\widehat{Q}_{x_1}$, and $\widehat{Q}_{x_0}$ using observational data $\{(X_i, Y_i, Z_i, W_i)\}_{i=1}^n$
3: **for** $t = 0, 1, \ldots, T-1$ **do**
4:     **for** $i = 1, \ldots, 4$ **do**
5:         **for** $j = 1, \ldots, 4$ **do**
6:             $\theta_{ij}^{(t)} \leftarrow \max\{\min\{\theta_{ij}^{(t)}, 1\}, 0\}$
7:             **if** $j > 2$ **then**
8:                 $\theta_{ij}^{(t)} \leftarrow \max\{\theta_{ij}^{(t)}, 1 - \sum_{j=2}^{j-1} \theta_{ij}^{(t)}\}$
9:             **end if**
10:         **end for**
11:     **end for**
12:     $\Theta^{(t+1)} \leftarrow \Theta^{(t)} - \alpha \frac{\partial}{\partial \Theta} L(\Theta; \widehat{P}_{x_1}, \widehat{P}_{x_0}, \widehat{Q}_{x_1}, \widehat{Q}_{x_0}, \mu^{(t)}, \boldsymbol{\lambda}_{x_1}^{(t)}, \boldsymbol{\lambda}_{x_0}^{(t)})|_{\Theta = \Theta^{(t)}}$
13:     $\mu^{(t+1)} = \mu^{(t)} + \rho(\mathbf{1}^\top (\widehat{P}_x(\Theta^{(t+1)})^{-1}))^\top \boldsymbol{e}_1 - 1)$
14:     **for** $j = 1, \ldots, 4$ **do**
15:         **if** $\lambda_{x_1,j}^{(t)} + \rho(\boldsymbol{e}_j^\top (\widehat{P}_{x_1}(\Theta^{(t+1)})^{-1}))^\top \boldsymbol{e}_1 - 1) > 1$ **then**
16:             $\lambda_{x_1,j}^{(t+1)} \leftarrow \lambda_{x_1,j}^{(t)} + \rho(\boldsymbol{e}_j^\top (\widehat{P}_{x_1}(\Theta^{(t+1)})^{-1})^\top \boldsymbol{e}_1 - 1)$
17:             $\lambda_{x_0,j}^{(t+1)} \leftarrow \lambda_{x_0,j}^{(t)} + \rho(\boldsymbol{e}_j^\top (\widehat{P}_{x_0}(\Theta^{(t+1)})^{-1})^\top \boldsymbol{e}_1 - 1)$
18:         **else if** $\lambda_{x,j}^{(t)} + \rho \boldsymbol{e}_j^\top (\widehat{P}_x(\Theta^{(t+1)})^{-1}))^\top \boldsymbol{e}_1 < 1$ **then**
19:             $\lambda_{x_1,j}^{(t+1)} \leftarrow \lambda_{x_1,j}^{(t)} + \rho \boldsymbol{e}_j^\top (\widehat{P}_{x_1}(\Theta^{(t+1)})^{-1})^\top \boldsymbol{e}_1$
20:             $\lambda_{x_0,j}^{(t+1)} \leftarrow \lambda_{x_0,j}^{(t)} + \rho \boldsymbol{e}_j^\top (\widehat{P}_{x_0}(\Theta^{(t+1)})^{-1})^\top \boldsymbol{e}_1$
21:         **else**
22:             $\lambda_{x_1,j}^{(t+1)} \leftarrow 0$; $\lambda_{x_0,j}^{(t+1)} \leftarrow 0$
23:         **end if**
24:     **end for**
25: **end for**
26: $\Pi^* \leftarrow \underset{\Pi_\Theta}{\arg\min} \Big\{ \|(\Pi\Theta^{(T)})\widehat{P}_{x_1}^{-1}\widehat{Q}_{x_1} - M_{x_1}(\Pi\Theta^{(T)})\|_F^2$
                $+ \|(\Pi\Theta^{(T)})\widehat{P}_{x_0}^{-1}\widehat{Q}_{x_0} - M_{x_0}(\Pi\Theta^{(T)})\|_F^2 \Big\}$
27: $\widehat{\Theta} \leftarrow \Pi^*\Theta^{(T)}$
28: $\widehat{\boldsymbol{u}} \leftarrow (\frac{1}{n}\sum_{i=1}^n \mathbf{1}\{X_i = x_1\})(\widehat{P}_{x_1}\widehat{\Theta}^{-1})^\top \boldsymbol{e}_1 + (\frac{1}{n}\sum_{i=1}^n \mathbf{1}\{X_i = x_0\})(\widehat{P}_{x_0}\widehat{\Theta}^{-1})^\top \boldsymbol{e}_1$

---

**Theorem B.1.** *Let*

$$F(\Theta; P_{x_1}, P_{x_0}, Q_{x_1}, Q_{x_0}) = \mathrm{vec}\left(\frac{\partial}{\partial\Theta} L(\Theta; P_{x_1}, P_{x_0}, Q_{x_1}, Q_{x_0}, \mu, \boldsymbol{\lambda}_{x_1}, \boldsymbol{\lambda}_{x_0})\right)$$

*and*

$$J = \frac{\partial F(\Theta; P_{x_1}, P_{x_0}, Q_{x_1}, Q_{x_0})}{\partial\,\mathrm{vec}(\Theta)^\top}, \quad K = \frac{\partial F(\Theta; P_{x_1}, P_{x_0}, Q_{x_1}, Q_{x_0})}{\partial\,\mathrm{vec}([P_{x_1}; P_{x_0}; Q_{x_1}; Q_{x_0}])^\top},$$

*where* $\mathrm{vec}(\cdot)$ *is a vec operator that transforms a matrix into a column vector by vertically stacking the columns of the matrix. For the asymptotic covariance matrix* $\Sigma$ *of* $\mathrm{vec}\left([\widehat{P}_{x_1}; \widehat{P}_{x_0}; \widehat{Q}_{x_1}; \widehat{Q}_{x_0}]\right)$ *and* $\widehat{\boldsymbol{u}} = \widehat{p}(x_1)(\widehat{P}_{x_1}\widehat{\Theta}^{-1})^\top \boldsymbol{e}_1 + \widehat{p}(x_0)(\widehat{P}_{x_0}\widehat{\Theta}^{-1})^\top \boldsymbol{e}_1$ *which is obtained by Algorithm 2, when both* $J$ *and* $J^{-1}K\Sigma K^\top J^{-\top}$ *are invertible, we have*

$$\sqrt{n}(\widehat{\boldsymbol{u}} - \boldsymbol{u}_0) \xrightarrow{d} \mathcal{N}(0, \Sigma_u)$$

*for* $\boldsymbol{u}_0 = p(x_1)(P_{x_1}\Theta_0^{-1})^\top \boldsymbol{e}_1 + p(x_0)(P_{x_0}\Theta_0^{-1})^\top \boldsymbol{e}_1$ *around* $\Theta_0$ *that is one of the solutions of* $F(\Theta; P_{x_1}, P_{x_0}, Q_{x_1}, Q_{x_0}) = 0$, *where*

$$\Sigma_u = \left[\left\{\boldsymbol{e}_1^\top \left(p(x_1)P_{x_1} + p(x_0)P_{x_0}\right)\right\} \otimes I\right]\left(J^{-1}K\Sigma K^\top J^{-\top}\right)^{-1}$$

$$\times \left[ \left\{ \boldsymbol{e}_1^\top \left( p(x_1)P_{x_1} + p(x_0)P_{x_0} \right) \right\} \otimes I \right]^\top$$

*and the notation "$-\top$" stands for a transposed inverse matrix.*

*Proof.* For $\widehat{\Theta}$ satisfying $F(\Theta; \widehat{P}_1, \widehat{P}_0, \widehat{Q}_1, \widehat{Q}_0, \mu, \boldsymbol{\lambda}_{x_1}, \boldsymbol{\lambda}_{x_0}) = 0$, from the Corollary 1 in Benichou and Gail [1], we have

$$\sqrt{n} \left( \text{vec}(\widehat{\Theta}) - \text{vec}(\Theta_0) \right) \xrightarrow{d} \mathcal{N} \left( 0, J^{-1}K\Sigma K^\top J^{-\top} \right). \tag{B.29}$$

Then, we have

$$\sqrt{n}(\widehat{\boldsymbol{u}} - \boldsymbol{u}_0)$$
$$= \sqrt{n} \, \text{vec} \left( \widehat{p}(x_1)(\widehat{P}_{x_1}\widehat{\Theta}^{-1})^\top \boldsymbol{e}_1 + \widehat{p}(x_0)(\widehat{P}_{x_0}\widehat{\Theta}^{-1})^\top \boldsymbol{e}_1 \right)$$
$$\quad - \sqrt{n} \, \text{vec} \left( p(x_1)(P_{x_1}\Theta_0^{-1})^\top \boldsymbol{e}_1 + p(x_0)(P_{x_0}\Theta_0^{-1})^\top \boldsymbol{e}_1 \right)$$
$$= \widehat{p}(x_1) \left\{ \left( \boldsymbol{e}_1^\top \widehat{P}_{x_1} \right) \otimes I \right\} \sqrt{n} \, \text{vec}(\widehat{\Theta}^{-\top}) + \widehat{p}(x_0) \left\{ \left( \boldsymbol{e}_1^\top \widehat{P}_{x_0} \right) \otimes I \right\} \sqrt{n} \, \text{vec}(\widehat{\Theta}^{-\top})$$
$$\quad - p(x_1) \left\{ \left( \boldsymbol{e}_1^\top P_{x_1} \right) \otimes I \right\} \sqrt{n} \, \text{vec}(\Theta_0^{-\top}) + p(x_0) \left\{ \left( \boldsymbol{e}_1^\top P_{x_0} \right) \otimes I \right\} \sqrt{n} \, \text{vec}(\Theta_0^{-\top})$$
$$= \left[ \widehat{p}(x_1) \left\{ \left( \boldsymbol{e}_1^\top \widehat{P}_{x_1} \right) \otimes I \right\} + \widehat{p}(x_0) \left( \boldsymbol{e}_1^\top \widehat{P}_{x_0} \otimes I \right) \right] \sqrt{n} \, \text{vec}(\widehat{\Theta}^{-\top})$$
$$\quad - \left[ p(x_1) \left\{ \left( \boldsymbol{e}_1^\top P_{x_1} \right) \otimes I \right\} + p(x_0) \left( \boldsymbol{e}_1^\top P_{x_0} \otimes I \right) \right] \sqrt{n} \, \text{vec}(\Theta_0^{-\top})$$
$$= \left[ \left\{ \boldsymbol{e}_1^\top \left( \widehat{p}(x_1)\widehat{P}_{x_1} + \widehat{p}(x_0)\widehat{P}_{x_0} \right) \right\} \otimes I \right] \sqrt{n} \, \text{vec}(\widehat{\Theta}^{-\top})$$
$$\quad - \left[ \left\{ \boldsymbol{e}_1^\top \left( p(x_1)P_{x_1} + p(x_0)P_{x_0} \right) \right\} \otimes I \right] \sqrt{n} \, \text{vec}(\Theta_0^{-\top})$$
$$= \left[ \left\{ \boldsymbol{e}_1^\top \left( \widehat{p}(x_1)\widehat{P}_{x_1} + \widehat{p}(x_0)\widehat{P}_{x_0} \right) \right\} \otimes I \right] \sqrt{n} \, \text{vec}(\widehat{\Theta}^{-\top} - \Theta_0^{-\top})$$
$$\quad + \left[ \left\{ \boldsymbol{e}_1^\top \left( \widehat{p}(x_1)\widehat{P}_{x_1} + \widehat{p}(x_0)\widehat{P}_{x_0} \right) \right\} \otimes I - \left\{ \boldsymbol{e}_1^\top \left( p(x_1)P_{x_1} + p(x_0)P_{x_0} \right) \right\} \otimes I \right] \sqrt{n} \, \text{vec}(\Theta_0^{-\top}),$$

where $\otimes$ stands for the Kronecker product. Then, from $\widehat{P}_x \xrightarrow{p} P_x$, $\widehat{p}(x) \xrightarrow{p} p(x)$ for $x \in \{x_1, x_0\}$, and (B.29), we have

$$\sqrt{n}(\widehat{\boldsymbol{u}} - \boldsymbol{u}_0) \xrightarrow{d} \mathcal{N}(0, \Sigma_u)$$

from Slutsky's lemma, where

$$\Sigma_u = \left[ \left\{ \boldsymbol{e}_1^\top \left( p(x_1)P_{x_1} + p(x_0)P_{x_0} \right) \right\} \otimes I \right] \left( J^{-1}K\Sigma K^\top J^{-\top} \right)^{-1}$$
$$\times \left[ \left\{ \boldsymbol{e}_1^\top \left( p(x_1)P_{x_1} + p(x_0)P_{x_0} \right) \right\} \otimes I \right]^\top.$$

$\square$

## C   Numerical Experiments

In this section, we investigate more properties of our proposed estimators through more numerical experiments in addition to Section 5. Letting $X, Y, Z, W$, and $U$ be discrete variables, we consider the causal diagrams shown in Fig. 2, where the joint probabilities of $(X, Y, Z, W, U)$ are given according Table C.1. Note that the distribution of $(p(u_1), p(u_2), p(u_3), p(u_4))$ is unbalanced differently from Section 5. Under the situation where $(X, Y, Z, W)$ can be observed but $U$ can not, the properties of the proposed estimators $\widehat{p}(u_2)$ and $\widehat{p}(u_2) - \widehat{p}(u_3)$ of $p(u_2)$ and $p(u_2) - p(u_3)$, respectively, are verified in the numerical experiments with sample sizes $n = 100, 200, 1000$, and $5000$.

Table C.2 and Fig. C.1 show the basic statistics and the box plots of $\widehat{p}(u_2)$ and $\widehat{p}(u_2) - \widehat{p}(u_3)$ for the above situations, respectively. The horizontal lines in Fig. C.1 show the true values of $p(u_2)$ and $p(u_2) - p(u_3)$. As seen from Table C.2, the sample means of $\widehat{p}(u_2)$ and $\widehat{p}(u_2) - \widehat{p}(u_3)$ are close to the true values and the sample standard deviations are smaller as the sample size is larger. Thus, it seems that the proposed estimation method provides the consistent estimators of $p(u_2)$ and $p(u_2) - p(u_3)$. From Fig. C.1, the interquantile ranges for $\widehat{p}(u_2)$ and

Table C.1: Conditional probability tables in another simulation.

| | (a) $p(Z\|U)$ | | | | (b) $p(W\|U)$ | | | | (c) $p(Y=1\|X,U)$ | |
|---|---|---|---|---|---|---|---|---|---|---|
| | $Z=1$ | $Z=2$ | $Z=3$ | $Z=4$ | $Z=1$ | $Z=2$ | $Z=3$ | $Z=4$ | $X=1$ | $X=0$ |
| $U=1$ | 7/10 | 1/10 | 1/10 | 1/10 | 7/10 | 1/10 | 1/10 | 1/10 | 1 | 1 |
| $U=2$ | 1/10 | 7/10 | 1/10 | 1/10 | 1/10 | 7/10 | 1/10 | 1/10 | 1 | 0 |
| $U=3$ | 1/10 | 1/10 | 7/10 | 1/10 | 1/10 | 1/10 | 7/10 | 1/10 | 0 | 1 |
| $U=4$ | 1/10 | 1/10 | 1/10 | 7/10 | 1/10 | 1/10 | 1/10 | 7/10 | 0 | 0 |

| | (d) $p(X=1\|W,U)$ | | | | (e) $p(U)$ |
|---|---|---|---|---|---|
| | $W=1$ | $W=2$ | $W=3$ | $W=4$ | |
| $U=1$ | 21/46 | 18/43 | 18/43 | 18/43 | 5/16 |
| $U=2$ | 9/34 | 21/71 | 9/34 | 9/34 | 5/16 |
| $U=3$ | 9/34 | 9/34 | 21/71 | 9/34 | 5/16 |
| $U=4$ | 9/34 | 9/34 | 9/34 | 21/71 | 1/16 |

Table C.2: Basic statistics in case that $\boldsymbol{u} = (5/16, 5/16, 5/16, 1/16)^\top$.

| | (a) $\widehat{p}(u_2)$ | | | | (b) $\widehat{p}(u_2) - \widehat{p}(u_3)$ | | | |
|---|---|---|---|---|---|---|---|---|
| | $n=100$ | $n=200$ | $n=1000$ | $n=5000$ | $n=100$ | $n=200$ | $n=1000$ | $n=5000$ |
| Minimum | 0.009 | 0.010 | 0.001 | 0.001 | $-0.942$ | $-0.946$ | $-0.961$ | $-0.894$ |
| 1st Quantile | 0.194 | 0.214 | 0.257 | 0.272 | $-0.120$ | $-0.093$ | $-0.064$ | $-0.060$ |
| Median | 0.256 | 0.268 | 0.285 | 0.300 | $-0.014$ | $-0.011$ | $-0.014$ | $-0.007$ |
| Mean | 0.259 | 0.270 | 0.284 | 0.299 | $-0.034$ | $-0.030$ | $-0.044$ | $-0.034$ |
| 3rd | 0.314 | 0.316 | 0.312 | 0.320 | 0.071 | 0.054 | 0.026 | 0.030 |
| Maximum | 0.895 | 0.900 | 0.853 | 0.872 | 0.866 | 0.888 | 0.802 | 0.796 |
| s.e. | 0.116 | 0.104 | 0.087 | 0.082 | 0.217 | 0.197 | 0.185 | 0.171 |

$\widehat{p}(u_2) - \widehat{p}(u_3)$ are narrower and still include the true values even if the sample size is large. In addition, the outliers would occur when it is difficult to judge that Condition 6 holds from observed data. Here, note that $\widehat{p}(u_2)$ may be underestiamted differently from Section 5. This is because $p(u_4)$ is truncated by zero for the finite sample size and $p(u_2)$ is greater than $p(u_4)$ in the setting.

## D  Case study

We illustrate our results through the data set reported by LaLonde [4] and re-analyzed by Dehejia and Wahba [3]. The aim of this study was to evaluate the effect on trainee earnings of the National Supported Work (NSW) demonstration, a job training program, in the field experiment. According to LaLonde [4], in this study, individuals were randomly assigned to treatment (attendance) and control groups (non-attendance) with the estimates that would have been produced by an econometrician, however it seem that the random assignment was not successful. The data set used in this section is available from Dehejia's homepage (`https://users.nber.org/~rdehejia/nswdata2.html`). The sample size given in the homepage is 445, and the variables of our interest are as follows:

$X$: an indicator for whether the individual attends the job training program ($x_1$: "attend"; $x_0$: "not attend"),

$Y$: an indicator for whether the individual's earning increment was increasing compared between 1975 and 1978 ($y_1$: "increasing"; $y_0$: "not increasing"),

$Z$: a joint indicator for marriage status and high school degree ($z_1$: non-zero earning in 1975 and "marriage"; $z_2$: non-zero earning in 1975 and "no marriage"; $z_3$: zero earning in 1975 and "marriage"; $z_4$: zero earning in 1975 and "no marriage"),

$W$: an indicator for age in years ($w_1$: age < 20; $w_2$: $20 \leq$ age < 27; $w_3$: $27 \leq$ age < 35; $w_4$: age $\geq$ 35).

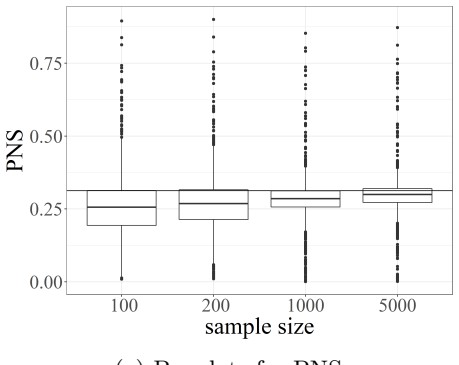
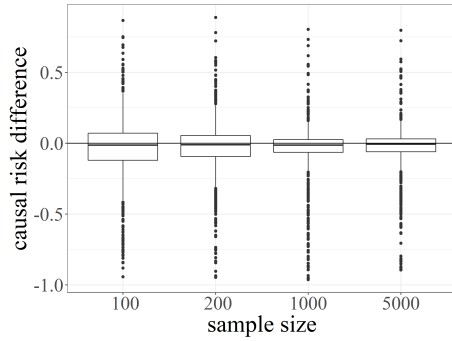

(a) Boxplots for PNS

(b) Boxplots for causal risk difference

Figure C.1: Boxplots of estimates based on the proposed method in case that $\boldsymbol{u} = (5/16, 5/16, 5/16, 1/16)$.

Table D.1: Estimates of PNS and causal risk difference in NSW dataset.

|  | Estimate (95%CI) |
| --- | --- |
| PNS | $0.297\ (0.041,\ 0.704)$ |
| causal risk difference | $0.061\ (-0.612,\ 0.615)$ |

Under the situation, we assume that the data generating process of this study is encoded in Figure 2.

In the data set, the sample estimations of $P_{x_1}$, $P_{x_0}$, $Q_{x_1}$, and $Q_{x_0}$ are given by

$$\widehat{P}_{x_1} = \begin{pmatrix} 1.000 & 0.141 & 0.568 & 0.049 \\ 0.205 & 0.011 & 0.189 & 0.000 \\ 0.416 & 0.054 & 0.195 & 0.022 \\ 0.254 & 0.038 & 0.135 & 0.022 \end{pmatrix}, \quad \widehat{P}_{x_0} = \begin{pmatrix} 1.000 & 0.115 & 0.719 & 0.038 \\ 0.262 & 0.008 & 0.242 & 0.000 \\ 0.404 & 0.050 & 0.288 & 0.008 \\ 0.242 & 0.054 & 0.112 & 0.023 \end{pmatrix},$$

$$\widehat{Q}_{x_1} = \begin{pmatrix} 0.670 & 0.103 & 0.351 & 0.032 \\ 0.146 & 0.011 & 0.130 & 0.000 \\ 0.276 & 0.038 & 0.119 & 0.016 \\ 0.162 & 0.032 & 0.076 & 0.011 \end{pmatrix}, \quad \widehat{Q}_{x_0} = \begin{pmatrix} 0.581 & 0.046 & 0.446 & 0.023 \\ 0.177 & 0.008 & 0.169 & 0.000 \\ 0.223 & 0.019 & 0.177 & 0.004 \\ 0.135 & 0.015 & 0.062 & 0.015 \end{pmatrix},$$

respectively. From these equations, it would be reasonable that Conditions 4 and 6 of Theorem 2 hold. Then, under the assumption that Condition 5 of Theorem 2 holds, together with Conditions 4 and 6, $p(u_1)$, $p(u_2)$, $p(u_3)$ and $p(u_4)$ are estimated as

$$\widehat{p}(u_1) = 0.289, \quad \widehat{p}(u_2) = 0.297, \quad \widehat{p}(u_3) = 0.236, \quad \widehat{p}(u_4) = 0.177,$$

through the proposed estimation method, respectively. From these probabilities, PNS $p(u_2)$ and the causal risk difference $p(u_2) - p(u_3)$ are evaluated by $\widehat{p}(u_2) = 0.297$ and $\widehat{p}(u_2) - \widehat{p}(u_3) = 0.061$, respectively. Table D.1 shows the estimates of PNS and causal risk difference with 95% confidential intervals. Here, the 2.5th and 97.5th percentiles of 1000 bootstrap replications of the estimates to derive the 95% confidential intervals[1].