# OpenReview forum: "Identification and Estimation of Joint Probabilities of Potential Outcomes in Observational Studies with Covariate Information"
_NeurIPS.cc/2021/Conference — NeurIPS 2021 Poster_

### Official Review · Reviewer_CSH4 · 2021-07-17

**Rating:** 7
**Confidence:** 4

**Summary:**

This paper proposes two nonstandard identification strategies for the joint distribution of counterfactuals. One is based on a single proxy $Z$ for the unknown confounder with at least $4$ levels, such that $p(y|do(z))$ is identifiable. The other one is based two proxies $Z$ and $W$ with at least $4$ levels for each. Both identification strategies are fully nonparametric. They are given by solutions of a complex system which are nonlinear in the observed distributions. Beyond identification, the paper also proposes estimation procedures and proves the asymptotic normality of the estimators. The estimation procedures are essentially the plug-in version of the system involved in the identification strategy.

**Limitations And Societal Impact:**

All my comments are minor.

(1) For case 1, don't you need $(X, Y)\perp Z | U$ instead of just $X\perp Z | U$?

(2) It would be more clear if $S$ and $M_x$ can be defined in Section 4. Otherwise, Section 4 is not readable enough. If the page limit is a concern, perhaps Section 3.1 can be trimmed a bit because the main text only discusses the estimation for Case 2.

(3) Suppose $Z$ (and $W$) have more than four levels, is it possible to combine all levels instead of just choosing three of them? A naive idea is to ensemble (9) with all subsets of three levels. But can we expand $P_x, Q_x$ by including all levels?

(4) Could you provide some heuristic explanation on how you get these strategies? More broadly, is it possible to get some generic principle from these ideas (like linear programming for other problems).

**Main Review:**

The identification strategies are mind-blowing. Most (partial) identification strategies are based on linear programming. But the proposed ones are certainly beyond that. I really enjoy the smart ideas. Furthermore, the assumptions for those two strategies appear to be reasonable in practice. Therefore, I think this work is both intellectually and practically interesting.

**Time Spent Reviewing:**

4

---

> ### Author Response · Authors · 2021-08-09
> **Response to Reviewer CSH4**
>
> > The identification strategies are mind-blowing. Most (partial) identification strategies are based on linear programming. But the proposed ones are certainly beyond that. I really enjoy the smart ideas. Furthermore, the assumptions for those two strategies appear to be reasonable in practice. Therefore, I think this work is both intellectually and practically interesting.
>
> Thank you for your positive evaluation.
> Here, we would like to reply to your comments in "Limitations And Societal Impact".
>
> **Q1** For case 1, don't you need $(X, Y) \perp Z|U$ instead of just $X\perp Z|U$?
>
> **A1** Thank you for making us aware of the typo. We will revise Condition 2 to $\\{ X, Y \\} \perp Z | U$.
>
> **Q2** It would be more clear if $S$ and $M_x$ can be defined in Section 4. Otherwise, Section 4 is not readable enough. If the page limit is a concern, perhaps Section 3.1 can be trimmed a bit because the main text only discusses the estimation for Case 2.
>
> **A2** Thank you for making us aware of the typos. Due to the lack of space, we did not provide the explicit expressions for $S$ and $M_x$ for $x \in \\{ x_0,x_1 \\}$ in Section 4 because six lines are necessary to present them. Instead of that, since they play an essential role in the proof of Theorem 1, we referred to equation (23) of the supplemental material the Supplemental Material A.2 (exactly to say, equations (A.23) and (A.24) of the Supplemental Material A.2 but not equation (23)). However, since we agree with your comments, we will trim Section 3.1 and define $S$ and $M_x$ within the main text.
>
> **Q3** Suppose $Z$  (and $W$) have more than four levels, is it possible to combine all levels instead of just choosing three of them? A naive idea is to the ensemble (9) with all subsets of three levels. But can we expand $P_x$, $Q_x$ by including all levels?
>
> **A3** Yes, it is not because the invertibilities of $P$, $P_{x_1}$ and $P_{x_0}$ are required.
>
> **Q4** Could you provide some heuristic explanation on how you get these strategies? More broadly, is it possible to get some generic principle from these ideas (like linear programming for other problems)?
>
> **A4** When the potential outcome types are considered as the values taken by a latent variable, according to [1],
> the idea of this paper is motivated by the fact that the matrix decomposition of observed probabilities is similar to the effect restoration method proposed by [2]. In addition, although $U$ can contain an uncertain number of covariates, we found that the state space of $U$ can always be divided into four states (potential outcome types) for binary exposure and outcome from [1].
>
>
> **Reference**
>
> [1] Alexander Balke & Judea Pearl (1997) Bounds on Treatment Effects from Studies with Imperfect Compliance, Journal of the American Statistical Association, 92:439, 1171-1176.
>
> [2] Manabu Kuroki, Judea Pearl (2014) Measurement bias and effect restoration in causal inference, Biometrika, 101(2), 423–437.

---

> > ### Comment · Reviewer_CSH4 · 2021-08-23
> > **Reply to the authors**
> >
> > I would like to thank the authors for their response. I think the paper is great. Point identification for the joint counterfactual distribution is rare and it is very useful to find special graph structures for which the point identification is possible. I will keep my positive score.

---

### Official Review · Reviewer_8HrH · 2021-07-17

**Rating:** 6
**Confidence:** 3

**Summary:**

This paper solves the identification of the probability of necessity and sufficiency in the case of unobserved confounder and the existence of proxy variable(s). Authors provide a novel formulation to solve the problem (presenting an unusual system of linear equations.


**Ethical Concerns:**

.

**Limitations And Societal Impact:**

.

**Main Review:**


quality
The paper is in good shape (see clarity)

originality
The paper provides an answer (under a certain condition) for an important causal question in an original way (although solving system of linear equations have bee investigated in different contexts)

clarity
The paper is in general pleasant to read but there are several places where improvements can be made.

For example, conditions for Theorem 1 are too shallowly explained. How should we interpret P or Q? Why do we have a marginal probability at the top row and leftmost column? Why there is no probability with x0 and y0 in P? etc. It seems that one can replace the top and the leftmost with those unused probabilities. But I am not sure. There is too much gap unexplained. The same applies to Theorem 2. Px1 and Px2 (which can be represented in a single expression with just x\in x1, x2). Is invertibility implies that proxies (i.e. covariates) Z and W contain sufficient information to relate U (similar to (Yx, Yx’)) and X?

In the estimation section, it is difficult to read through lines 274—282, which refer to the results in supplementary materials extensively.

What is a singular model?

Empirical results section can be reduced (with Table 2 in Appendix). What can we do with outliers when Condition 6 doesn’t hold (or the matrix is close to singular)? It would be nice to plot invertibility test results together with the estimates (PNS and causal risk difference) to ensure that those outliers are mainly due to near-singular matrices.


significance
In the context of the theory of causal effect identifiability, this paper provides a novel identification result where non-trivially constructed matrices and their inverse is used.


limitation
Although some of the reviewers may point out the stringent assumptions this paper relies on. I do not consider them as the limitation but the ones that ‘enable’ the identification.


minor
As far as I know, the paper needs to use the basic font provided by NeurIPS.
Line 227, ito → it

**Time Spent Reviewing:**

6

---

> ### Author Response · Authors · 2021-08-09
> **Response to Reviewer 8HrH**
>
> > Quality: The paper is in good shape (see clarity)
> >
> > Originality: The paper provides an answer (under a certain condition) for an important causal question in an original way (although the solving systems of linear equations have been investigated in different contexts)
>
> Thank you for your positive evaluation.
>
> > Clarity: The paper is in general pleasant to read but there are several places where improvements can be made.
> For example, conditions for Theorem 1 are too shallowly explained. How should we interpret $P$ or $Q$?
>
> Thank you for your comments.
> For Theorem 1, $P$ and $Q$ are given in equations (2) and (3), respectively.
> In addition, for Theorem 2, $P_{x_1}$ and $P_{x_0}$ are given in equations (5) and (6), respectively.
> We have introduced $P$, $Q$, $P_{x_0}$ and $P_{x_1}$ as mathematical tools to derive Theorems 1 and 2 according to the idea of [1].
>
> > Why do we have a marginal probability at the top row and leftmost column?
> Why is there no probability with $x_0$ and $y_0$ in $P$?
>
> Marginal probabilities appear at the top row and leftmost column of $P$ and $Q$ from the correspondence between the matrix decomposition and the law of total probability under the assumptions.
> By the similar reason above, the probabilities of $(x_0,y_0)$ do not appear in $P$; such information can be recovered from $1$, $p(x_1,y_1)$, $p(x_0,y_1)$ and $p(x_1,y_0)$.
>
> > It seems that one can replace the top and the leftmost with those unused probabilities. But I am not sure. There is too much gap unexplained. The same applies to Theorem 2. $P_{x_1}$ and $P_{x_2}$ (which can be represented in a single expression with just $x\in \\{x_1, x_2\\}$).
>
> We also had the same thought as the reviewer when we started this research.
> However, we found that it is more complicated to derive the results of the paper if the "unused probabilities" are used.
> Here, we will represent $P_{x_1}$ and $P_{x_2}$ in a single expression with $x\in \\{x_1, x_2\\}$.
>
> > Is invertibility implies that proxies  (i.e., covariates) $Z$ and $W$ contain sufficient information to relate $U$ (similar to $(Y_x, Y_{x'})$) and $X$?
>
> No, it is not.
> The invertibility of $P$ implies that the proxies contain "necessary" information but not "sufficient" information to relate $U$ (similar to $(Y_x, Y_{x'}))$ and $X$ .
>
> > In the estimation section, it is difficult to read through lines 274-282, which refer to the results in supplementary materials extensively.
>
> The central aim of the paper was to solve the identification problem because the proposed estimation method is meaningless if the joint probabilities of potential outcomes are not identifiable.
> Then, because of the page limitation, we had no choice but to move the details on the estimation to the Supplemental Material.
>
> > What is a singular model?
>
> A statistical model is regular if it is identifiable and the Fisher information matrix is positive definite. A model is singular if it is not regular.
> Since it is known that many latent variable models are singular, we use the word without definition.
> Within the page limit, we will try to add the definition in the main text.
>
> > Empirical results section can be reduced (with Table 2 in Appendix). What can we do with outliers when Condition 6 does not hold (or the matrix is close to singular)?
> It would be nice to plot invertibility test results together with the estimates (PNS and causal risk difference) to ensure that those outliers are mainly due to near-singular matrices.
>
> Thank you for your insightful comments. When Condition 6 does not hold, the joint probabilities of potential outcomes are not estimable by the proposed estimation method.
> In addition, theoretically, as far as the matrix is not singular, Theorem 2 shows that the joint probabilities of potential outcomes are estimable.
> Thus, we did not consider the outlier problem of observed values since we use joint probabilities from a single dataset to estimate the joint probabilities of potential outcomes.
> However, as you stated, since it is very important to know the correspondence between the estimation accuracy and the singular model regarding our research,
> we will state the outlier problem as the future work in the main text, together with your comment "to plot ... matrices", within the page limit.
>
> > Significance: In the context of the theory of causal effect identifiability, this paper provides a novel identification result where non-trivially constructed matrices and their inverse is used.
>
> Thank you for your comments.
>
> > Limitation: Although some of the reviewers may point out the stringent assumptions this paper relies on. I do not consider them as the limitation but the ones that ``enable" the identification.
>
> Thank you for your comments that support our results.
>
> > Minor: As far as I know, the paper needs to use the basic font provided by NeurIPS.
> Line 227, ito → it
>
> We will revise the paper.
>
>
> **Reference**
>
> [1] Manabu Kuroki, Judea Pearl (2014) Measurement bias and effect restoration in causal inference, Biometrika, Volume 101(2), 423–437.

---

> > ### Comment · Reviewer_8HrH · 2021-08-21
> > **follow up**
> >
> > Thanks for the response.
> >
> > I didn’t mean to ask the “literal” meaning of P or Q. The paper defines P and Q,  and Theorem 1 and conditions are clearly written, mathematically. The problem is that no one will ever understand why these conditions lead to the identifiability without looking at the Appendix. No intuitive connection has been made between matrices and the identifiability in the main text.

---

> > > ### Author Response · Authors · 2021-08-22
> > > **Thank you for the follow-up**
> > >
> > > Thank you for your comments.
> > >
> > > The derivations of this paper are inspired by the proofs of the identification condition for latent class models by, for example, [1] and [2] and the effect restoration by [3]. However, as far as we know, they did not provide the intuitive meaning of such matrices. Thus, it is difficult for us, and it may have also been difficult for previous researchers to provide an intuition for the matrices $P$ and $Q$ as you say.
> > >
> > > Here, although it is based on the form of the explanation of Theorems, we are considering adding the explanations of $P$ and $Q$ in main text within the page limit, referring to our response to the comment from reviewer 7Tux "It would be nice to .... Conditions 1-3, and Conditions 4-6 could hold in practice."
> > >
> > > **Reference**
> > >
> > > [1] T. W. Anderson (1954). On estimation of parameters in latent structure analysis. Psychometrika 19, 1-10.
> > >
> > > [2] W. A.  Gibson (1955). An extension of Anderson's solution for the latent structure equations. Psychometrika 20, 69-73.
> > >
> > > [3] M. Kuroki & J. Pearl (2014). Effect restoration and measurement bias in causal inference, Biometrika, 101, 423-437.

---

> > > > ### Comment · Reviewer_8HrH · 2021-09-02
> > > > **follow-up**
> > > >
> > > > I maintain my positive assessment.

---

### Official Review · Reviewer_Mum6 · 2021-07-22

**Rating:** 6
**Confidence:** 4

**Summary:**

In this paper, authors propose a mechanism to calculate joint probabilities of potential outcome under the assumption that certain covariate information is available. The first part of the paper identifies conditions under which these probabilities can be estimated and the second part aims to use augmented Lagrangian method to calculate the probability.



**Limitations And Societal Impact:**

None identified

**Main Review:**

The problem of relaxing assumptions of prior techniques would be an impactful contribution for the literature on explainable AI.

Weak points:
1. Motivation: It is unclear how the presented techniques are applicable in practice. For example, the covariate assumption is not available in most real-world settings.

2. Theory: It seems like the conditional independence assumption X\indep Z | U makes the analysis straightforward. I am not sure how the analysis would extend if this condition does not hold.

3. The intuition behind the proposed method in section 4 is unclear. Please add more details and discuss the proof techniques.

4. The proof is discussed with respect to two cases. How does it generalize it to other graph structures?

5.  Evaluation: The empirical evaluation of the presented techniques is very superficial.
a) The experiment is conducted on a very small synthetic causal graph. Please consider a real-world dataset that motivates the motivation and consider more than 4 variables.
b) There is no comparison with prior techniques that make monotonicity assumptions. It is unclear if the proposed method really performs better.
c) In Figure 3, it maybe better to compare PNS scores of different attributes and validate its correctness with respect to what prior techniques report.

6. Paper is not well written. The notation is not defined clearly which makes it very hard to validate the correctness of some results.

7. Also, certain claims about medical results, e.g. lines 187-192 are not well justified. Please update the introduction to discuss these use cases and define formal notation.


**Time Spent Reviewing:**

4

---

> ### Author Response · Authors · 2021-08-09
> **Response to Reviewer Mum6**
>
> > The problem of relaxing assumptions of prior techniques would be an impactful contribution for the literature on explainable AI.
>
> Thank you for your positive evaluation.  Here, we would like to reply to your comments in "Main Review".
>
> **Q1** Motivation: It is unclear how the presented techniques are applicable in practice. For example, the covariate assumption is not available in most real-world settings.
>
> **A1** As we stated in Section 3.1, $U$ is considered the set of all discrete and continuous covariates that could affect $X$ and $Y$, both observed and unobserved, according to [1]. Thus $U$ can include both endogenous and exogenous variables as long as $U$ is not affected by $X$ or $Y$.
> In addition, although $U$ can contain an uncertain number of covariates,
> we found that the state space of $U$ can always be divided into four states (potential outcome types) for binary exposure and outcome from [1].
> Taking this into account, in our opinion, the assumption on covariates $U$, i.e., $Z\perp \\{ X, Y \\}|U$ ($Z \perp X | U$ in the paper was a typo), is not restricted in practice.
> Here, as we stated in Section 3, the statistical independence relationships among observed variables may depend on partitioning the states of $U$.
>
> **Q2** Theory: It seems like the conditional independence assumption $X\perp Z|U$ makes the analysis straightforward. I am not sure how the analysis would extend if this condition does not hold.
>
> **Q4** The proof is discussed with respect to two cases. How does it generalize it to other graph structures?
>
> **A2 and A4**
> As a direct extension of this paper, for Theorem 1, it is easy to show that the assumption of $\\{ X, Y \\} \perp Z|U$ can be relaxed to the assumption of $\\{ X, Y \\} \perp Z|\\{U, S\\}$ for a set of observed variables $S$.
> Then, $Z$ is not allowed to connect with $X$ and $Y$ given $U$ in the results of the paper, but $Z$ can be associated with $X$ through $S$ given $U$ under the assumption of $\\{ X, Y \\}\perp Z|\{U, S\}$.
> The same extension can be achieved for Theorem 2.
> Since these extensions can be proved in a similar way to these of Theorems 1 and 2, we do not state these extensions.
>
> **Q3** The intuition behind the proposed method in section 4 is unclear. Please add more details and discuss the proof techniques.
>
> **A3** The estimation strategy is motivated by the augmented Lagrangians.
> Although we stated the theoretical details and the proposed algorithm in the Supplemental Material B, we can add more explanations and discuss the proof techniques in the Supplemental Material if necessary.
>
> **Q5a** The experiment is conducted on a very small synthetic causal graph. Please consider a real-world dataset that motivates the motivation and consider more than 4 variables.
>
> **A5a** We can give a real-world example from [3] in the Supplement Material.
> Here, when there are many covariates, one could resort to the dimension reduction method which maps the cells of the proxy variables more than four variables onto a single scalar.
> Since the proposed assumption $Z \perp W|U$ is invariant in such a dimension reduction from the elementary theorem in statistics (if $Z \perp W|U$ holds, then $g(Z) \perp h(W)|U$ holds for the transformations $g$ and $h$), it is unnecessary to consider more than four variables.
>
> **Q5b** There is no comparison with prior techniques that make monotonicity assumptions. It is unclear if the proposed method really performs better.
>
> **A5b** As we stated in Section 3.1, the results of the paper are not applicable under the monotonicity assumption.
> Thus, it is necessary to reformulate our results to apply our results to cases where the monotonicity can be assumed.
> We did not provide such reformulations because they can be derived in a similar way to Theorems 1 and 2.
>
> **Q5c** In Figure 3, it may be better to compare PNS scores of different attributes and validate its correctness with respect to what prior techniques report.
>
> **A5c** Most of the prior techniques are applicable under the monotonicity assumption, and the bounding methods have been utilized when the assumption is violated.
> In contrast, the results of the paper identify the joint probabilities of potential outcomes without the monotonicity assumption but are not applicable under the monotonicity.
> Thus, since the applicable situations are separated between the prior and the proposed techniques, we did not compare the proposed method with prior techniques.
> However, we can add the numerical experiments comparing the results of the paper with the prior techniques with/without the monotonicity if necessary.
>
> **Q6** Paper is not well written. The notation is not defined clearly which makes it very hard to validate the correctness of some results.
>
> **A6** Thank you for your comments. We will revise the paper.
>
> **Q7** Also, certain claims about medical results, e.g., lines 187-192, are not well justified. Please update the introduction to discuss these use cases and define formal notation.
>
> **A7** We will update the introduction of the paper within the page limit.
>
>
> **Reference**
>
> [1] Alexander Balke & Judea Pearl (1997) Bounds on Treatment Effects from Studies with Imperfect Compliance, Journal of the American Statistical Association, 92:439, 1171-1176.
>
> [2] Robert J. LaLonde (1986) Evaluating the Econometric Evaluations of Training Programs with Experimental Data. The American Economic Review, 76(4), 604-620.

---

### Official Review · Reviewer_7Tux · 2021-07-30

**Rating:** 6
**Confidence:** 2

**Summary:**

This paper studies the identification and estimation problem of the joint probabilities of potential outcomes. To solve this problem, this paper proposes two sets of identification conditions using covariate information. Furthermore, this paper proposes a novel statistical estimation method based on the augmented Lagrangian method to evaluate the joint probabilities of potential outcomes under the proposed identification conditions.

**Limitations And Societal Impact:**

Identification with more than two covariates: The authors provide two sets of identification conditions. One is based on one covariate with causal risk, and the other one is based on two covariates without causal risk. What happens if there are more than two covariates, and what happens if there is more than one covariate with causal risk? It is worth explaining how the identification conditions can be generalized to these more general cases if these conditions are generalizable. Otherwise, it is also worth clarifying why these conditions are not generalizable.

Numerical simulation, practical relevance, and presentation of this paper: The authors mention some healthcare examples in the introduction, which are helpful to motivate the problem. If the authors could explain and interpret Theorem 1, Conditions 1-3, Theorem 2, and Conditions 4-6 using the examples in the introduction, the identification section will be much easier to read. In addition, this paper could have a broader impact if the authors could demonstrate the practical relevance of their methods through experiments on real data sets (e.g. the real data related to the heathcare examples mentioned in the introduction).


**Main Review:**

This paper studies the problem of evaluating the joint probabilities, which is important in causal inference for two reasons. First, if the joint probabilities are identifiable, then the causal risk is also identifiable. Second, the joint probabilities are critical to evaluate the "Probability of Sufficiency", “Probability of Necessity”, and “Probability of Necessity and Sufficiency”.

The two sets of identification conditions for the joint probabilities seem to be novel. However, I remain concerned about the practicality of these conditions. It would be nice to give a few examples to illustrate when Conditions 1-3, and Conditions 4-6 could hold in practice. Moreover, it will be helpful to clarify the conceptual challenges in proposing these conditions.

The estimation approach based on the augmented Lagrangian method is a nice solution to estimate the joint probabilities, given that the standard statistical estimation methods do not work. I appreciate that the authors provide theoretical guarantees for their estimation approach. It would be nice if the authors could provide a formal and rigorous statement of their theoretical results (e.g. a theorem or proposition) with the clearly stated assumptions (as opposed to just a sentence in line 286 and a section (i.e. B.3) in the appendix to show the proof).

Regarding numerical simulations, I appreciate that the desired properties are verified through simulations, but I remain concerned about the simulation setup. It would be nice to clarify why the numbers chosen in Table 1 are of practical relevance. In addition, it would be nice if the authors could demonstrate the performance of their estimation methods on real data sets.


**Time Spent Reviewing:**

5

---

> ### Author Response · Authors · 2021-08-09
> **Response to Reviewer 7Tux**
>
> > This paper studies the problem of evaluating the joint probabilities, which is important in causal inference for two reasons. First, if the joint probabilities are identifiable, then the causal risk is also identifiable. Second, the joint probabilities are critical to evaluate the Probability of Sufficiency, Probability of Necessity, and Probability of Necessity and Sufficiency.
>
> Thank you for your comments.
>
> > The two sets of identification conditions for the joint probabilities seem to be novel. However, I remain concerned about the practicality of these conditions. It would be nice to give a few examples to illustrate when Conditions 1-3, and Conditions 4-6 could hold in practice.
>
> First, Conditions 1 and 4 show situations where we can estimate the probabilities as $p(Y_{x_1}=y,Y_{x_0}=y')$ and $p(x_1, y_1, z, w)$ from the available data.
>
> Second, regarding Conditions 2 and 5, as we stated in Section 3.1, $U$ is considered the set of all discrete and continuous covariates that could affect $X$ and $Y$, both observed and unobserved, according to [1]. Thus $U$ can include both endogenous and exogenous variables as long as $U$ is not affected by $X$ or $Y$.
> Here, although $U$ can contain an uncertain number of covariates,
> we found that the state space of $U$ can always be divided into four states (potential outcome types) for binary exposure and outcome from [1].
> Taking this into account, in our opinion, the assumption on covariates $U$, i.e., $Z \perp \\{ X, Y \\}|U$ ($Z \perp X | U$ in the paper was a typo), is not restricted in practice.
> Here, as we stated in Section 3, the statistical independence relationships among observed variables may depend on partitioning the states of $U$.
>
> Third, Conditions 3 and 6 imply that (i) the problems of empty cells do not occur and (ii) variables of interest are not independent of each other, which would be reasonable assumptions in practical situations.
>
> > Moreover, it will be helpful to clarify the conceptual challenges in proposing these conditions.
>
> Initially, the conceptual challenge of the paper was to provide a counterexample to "sensitivity to the generative process" ([3], pp.284-285),
> "even in the absence of confounding, probabilities of certain counterfactual relationships cannot be identified from frequency information unless we specify the functional relationships that connect causes and effects. The functional specification is needed whenever the facts at hand (e.g., disease) might be affected by the counterfactual antecedent (e.g., exposure)".
> Intuitively, the results of the paper assume that (i) observed proxy variables have no direct association with $\\{X, Y\\}$ and (ii) $X$ can not be affected by $Y$. However, different from [3] and [4], we solved the identification problem without the assumptions of the monotonicity or the full functional specification of the structural causal model.
>
> > The estimation approach based on the augmented Lagrangian method is a nice solution to estimate the joint probabilities, given that the standard statistical estimation methods do not work. I appreciate that the authors provide theoretical guarantees for their estimation approach.
>
> Since it is complicated to show the asymptotic normality of the proposed estimating method within the main paper, we provided the proof in the Supplemental Material B.3.
>
> >It would be nice if the authors could provide a formal and rigorous statement of their theoretical results (e.g., a theorem or proposition) with the clearly stated assumptions (as opposed to just a sentence in line 286 and a section (i.e., B.3) in the appendix to show the proof).
>
> Within the page limit, we will provide the statement of the theoretical results in the main text.
>
> > Regarding numerical simulations, I appreciate that the desired properties are verified through simulations, but I remain concerned about the simulation setup. It would be nice to clarify why the numbers chosen in Table 1 are of practical relevance. In addition, it would be nice if the authors could demonstrate the performance of their estimation methods on real data sets.
>
> We provided Table 1 as a simple situation where the distribution $(p(u_1), p(u_2), p(u_3), p(u_4))$ is balanced, which would be a realistic simulation setting.
> In addition, as the different case from Table 1,
> we provided numerical examples where the distribution $(p(u_1), p(u_2), p(u_3), p(u_4))$ is unbalanced in Supplemental Material C.
> Here, we can provide a real-world example from [2] in the Supplement Material if the paper is accepted.
>
> > Identification with more than two covariates: The authors provide two sets of identification conditions. One is based on one covariate with causal risk, and the other one is based on two covariates without causal risk. What happens if there are more than two covariates, and what happens if there is more than one covariate with causal risk? It is worth explaining how the identification conditions can be generalized to these more general cases if these conditions are generalizable. Otherwise, it is also worth clarifying why these conditions are not generalizable.
>
> When there are many covariates, the problems of empty cells would occur.
> The results of the paper do not permit us to deal with such problems, owing to the high dimensionality of the proxy variables would prevent us from getting reliable statistics of the observed probabilities.
> To solve the problem, one could resort to the dimension reduction method which maps the cells of the proxy variables onto a single scalar.
> Since the proposed assumption $Z \perp W|U$ is invariant in such a dimension reduction from the elementary theorem in statistics (
> i.e., if $Z \perp W|U$ holds, then $g(Z) \perp h(W)|U$ holds for the transformations $g$ and $h$), we did not discuss the problem of many covariates in this paper.
>
> > Numerical simulation, practical relevance, and presentation of this paper: The authors mention some healthcare examples in the introduction, which are helpful to motivate the problem. If the authors could explain and interpret Theorem 1, Conditions 1-3, Theorem 2, and Conditions 4-6 using the examples in the introduction, the identification section will be much easier to read.
>
> Due to the page limitation, it may be difficult to explain Theorems 1 and 2 using the examples in Section 1. However, we will do that within the page limit.
>
> > In addition, this paper could have a broader impact if the authors could demonstrate the practical relevance of their methods through experiments on real data sets (e.g., the real data related to the healthcare examples mentioned in the introduction).
>
> Although the observational studies with covariate information are really common in health care science, as far as we know, the available data sets for us are restricted into one or two covariates with two or three categories at most.
> Instead, we can give a real-world example from [2] in the Supplemental Material if the paper is accepted.
>
>
> **Reference**
>
> [1] Alexander Balke & Judea Pearl (1997) Bounds on Treatment Effects from Studies with Imperfect Compliance, Journal of the American Statistical Association, 92:439, 1171-1176.
>
> [2] Robert J. LaLonde (1986) Evaluating the Econometric Evaluations of Training Programs with Experimental Data. The American Economic Review, 76(4), 604-620.
>
> [3] Judea Pearl (2009) Causality: Models, Reasoning and Inference. Cambridge University Press, 2nd edition.
>
> [4] Jin Tian & Judea Pearl (2000) Probabilities of causation: Bounds and identification. Annals of Mathematics and Artificial Intelligence 28, 287–313.

---

> > ### Comment · Reviewer_7Tux · 2021-09-03
> > **follow up**
> >
> > Thank the authors for the response and clarification. The response alleviates most of my concerns and I raised my score assuming the authors will, as promised, add
> > (1) a formal statement (e.g. a theorem with clearly stated assumptions) of the asymptotic normality results
> > (2) an empirical example from LaLonde (1986) to demonstrate the effectiveness of their method
> > (3) some short explanation of Theorems 1 and 2 using the examples in Section 1

---

### Decision · Program_Chairs · 2021-09-27

**Decision:**

Accept (Poster)

**Comment:**

All reviewers have a favorable opinion about this paper after discussion.

Paraphrasing one of the reviewer comments, identification results for joint probabilities of potential outcomes are "very rare" and definitely crucial for many important quantities like probability of sufficiency, necessity etc., individual treatment effect. This paper provides a novel conditions and an estimation procedure for solving it.

So clearly the contributions are fundamental, original and timely.


There is a reviewer comment asking for formal statement for the asymptotic normality, some empirical comparison and further examples for the theorems. I really hope authors can revise accordingly.